# Beyond Scale: The Diversity Coefficient as a Data Quality Metric for Variability in Natural Language Data

## Abstract

Current trends in pre-training Large Language Models (LLMs) primarily focus on the scaling of model and dataset size. While the *quality* of pre-training data is considered an important factor for training powerful LLMs, it remains a nebulous concept that has not been rigorously characterized. To this end, we propose a formalization of one key aspect of data quality – measuring the *variability* of natural language data – specifically via a measure we call the diversity coefficient. Our empirical analysis shows that the proposed diversity coefficient aligns with the intuitive properties of diversity and variability, e.g., it increases as the number of latent concepts increases. Then, we measure the diversity coefficient of publicly available pre-training datasets and demonstrate that their formal diversity is high compared to theoretical lower and upper bounds. Finally, we conduct a comprehensive set of controlled *interventional* experiments with GPT-2 and LLaMAv2 that demonstrate the diversity coefficient of pre-training data characterizes useful aspects of downstream model evaluation performance—totaling 44 models of various sizes (51M to 7B parameters). We conclude that our formal notion of diversity is an important aspect of data quality that captures variability and causally leads to improved evaluation performance.

## 1 Introduction

Current trends in pre-training Large Language Models (LLMs) tend to concentrate on model and dataset size scaling Chowdhery et al. (2022); Nostalgebraist (2022); OpenAI (2023); Google (2023). Therefore, vast amounts of effort have been invested in understanding neural scaling laws—the power-law relationship between the loss of artificial deep networks and the *size* of the pre-training dataset for a fixed compute budget (Hestness et al., 2017; Rosenfeld et al., 2019; Henighan et al., 2020; Kaplan et al., 2020; Gordon et al., 2021; Hernandez et al., 2021; Jones, 2021; Zhai et al., 2022; Hoffmann et al., 2022; Clark et al., 2022; Neumann & Gros, 2022). In addition, recent work focuses on training a fixed-size model but using very large, trillion-token datasets (Touvron et al., 2023a;b). However, the effectiveness of these systems also fundamentally relies on the quality Longpre et al. (2023), variability and coverage of the pre-training data Hashimoto (2021); David et al. (2010) and not only the *size*. In particular, the richness and variety of data, otherwise known as data diversity, is a key aspect of data quality that plays an important role in researchers' and practitioners' choice of pre-training corpora for general capabilities (Gao et al., 2020; Brown et al., 2020; Touvron et al., 2023a; Eldan & Li, 2023; Gunasekar et al., 2023). In other words, diverse data is high quality data when your goal is to instill *general capabilities* in a model.

In addition, experimental settings demonstrate that the level of variety and diversity in pre-training data is likely a strong causal factor in the development of in-context learning (ICL) in LLMs, an essential component of the versatility and generality of LLMs (Xie et al., 2022; Shin et al., 2022; Chan et al., 2022). However, data quality, diversity and coverage David et al. (2010) are often discussed in vague and imprecise ways Longpre et al. (2023). Even in instances of quantitative approaches to data quality, methods are often not interpretable as to *why* certain datasets are preferable over others, or require reference corpora and, thereby, limit the diversity of resulting datasets (Xie et al., 2023a;c). Hence, we propose to ground the discussion of data quality and, in particular, data diversity, through the Task2Vec *diversity coefficient for natural language*—a metric using the

expected distance between Task2Vec embeddings of data batches (Achille et al., 2019a; Miranda et al., 2022a) to quantify the level of structural and semantic diversity of natural language data, allowing researchers and practitioners to move beyond scale alone.

Hence, our key **contributions** are:

1. *A paradigm shift beyond dataset scale* to a data-centric machine learning perspective through a quantitative data diversity metric for data variability—the **diversity coefficient**.

2. We advance discussions on data quality by developing an interpretable and formal quantitative measure of an important aspect of data quality—data diversity—in the form of the Task2Vec *diversity coefficient for natural language*.

3. We demonstrate that the diversity coefficient aligns with human intuitions about variability and diversity through interpretability and relationship analyses. For example, if the number of latent concepts increases, a richer vocabulary is used, or datasets are concatenated, then the diversity coefficient should increase.

4. We quantitatively demonstrate that public datasets for LLM pre-training exhibit *high* diversity, by them to well-motivated lower and upper bounds for our diversity metric.

5. We provide evidence via **interventional** experiments with various GPT-2 and LLaMAv2 models that pre-training models on data with higher diversity coefficients leads to better downstream performance on language modeling for diverse evaluation corpora—totaling 44 models pre-trained from scratch of various sizes (54M to 7B).

We conclude that the diversity coefficient serves as an effective measure of the diversity of natural language data and, thus, enables researchers and practitioners to more rigorously study this aspect of natural language data quality and *move beyond scale alone*.

## 2 METHOD: THE DIVERSITY COEFFICIENT FOR NATURAL LANGUAGE

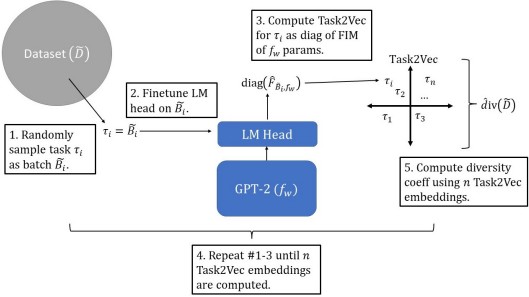

Figure 1: **The process of computing the diversity coefficient for a dataset proceeds through three main stages:** (a) randomly sampling batches of text from the dataset, (b) computing the Task2Vec embeddings for each sampled batch, and (c) calculating the expected pairwise cosine distance between the Task2Vec embeddings of the sampled data.

Our aim is to formalize the concept of data diversity and empirically demonstrate, through *interventional* experiments, its positive relationship with the evaluation performance of the validation set. We estimate the diversity of a dataset through the Task2Vec diversity coefficient for natural language, which embeds randomly sampled batches of data using Task2Vec embeddings (Achille et al., 2019a) and computes the expected cosine distance between these (see Figure 1). It approximates the diversity of the dataset because distances between Task2Vec embeddings approximate the distance between the generative (task) distributions of the batches.

### 2.1 COMPUTING TASK2VEC EMBEDDINGS FOR TEXT

Task2Vec embeddings are defined as the diagonal of an (approximated) Fischer Information Matrix (FIM); the FIM is derived from the parameters of a fixed probe network—in the case of text data, a language model—whose final layer has been fine-tuned on the target data to embed (Achille et al.,

2019a). Since the FIM encodes information on which parameters of the probe network are most important in predicting the text sequence, the diagonal of the FIM encodes a compact and tractable representation of these parameters. Thus, the Task2Vec embedding of text data represents which parameters of the probe network are most important in *solving* the next token prediction task given by a text sequence. We use a probe network with final layer fine-tuning on the target data in calculating the FIM as this is the only validated method (to our knowledge) for creating Task2Vec embeddings (Achille et al., 2019a). Importantly, the probe network is fixed for all batches in order to make embeddings comparable between tasks (Achille et al., 2019a).

Though Task2Vec embeddings are primarily used in visual and meta-learning tasks (Achille et al., 2019a; Miranda et al., 2022a; Wallace et al., 2021), we adapt and repurpose this technique to the novel domain of natural language data. In particular, we use GPT-2 (Radford et al., 2019) as our fixed probe network and fine-tune its final layer with a next-token prediction objective.

Formally, we compute the FIM as:

$$\hat{F}_B = \mathbb{E}_{x,t,\hat{x}_t} \nabla_w \log \hat{p}_w(\hat{x}_t | x_{t-1:1}) \nabla_w \log \hat{p}_w(\hat{x}_t | x_{t-1:1})^\top$$

The Task2Vec embedding $\vec{f}_B$ is the diagonal of the computed FIM, i.e. $\vec{f}_B = Diag(F_B)$, where $B$ represents a batch of text sequences, $x$ is a text sequence sampled from $B$, $\hat{x}_t$ is the next token predicted by the fine-tuned probe network $f_w$ (with weights $w$) conditioned on the ground truth sequence $x_{t-1:1}$, and $t$ is a given index of the sequence.

## 2.2 Computing the Diversity Coefficient for Natural Language Text

The *diversity coefficient* is the expected pairwise cosine distance $d$ between Task2Vec embeddings of batches of text $B_1, B_2$ sampled from a natural language dataset $D$:

$$\hat{div}(D) = \mathbb{E}_{B_1,B_2 \sim D} d(\vec{f}_{B_1}, \vec{f}_{B_2})$$

By measuring the distance between Task2Vec embeddings of sampled text batches, the diversity coefficient captures the average intrinsic variability of the sampled batches, serving as an efficient estimate for the diversity of information contained in the dataset. It is an effective "soft" count of the number of distinct batches.

A useful variation on the concept of the diversity coefficient is the *cross diversity coefficient*, which calculates the informational variation i.e. diversity *between separate* datasets. Following the framework of the diversity coefficient, the cross diversity coefficient is the expected pairwise cosine distance $d$ between Task2Vec embeddings of batches of text $B_1$ sampled from a natural language dataset $D_1$ and batches of text $B_2$ sampled from a separate dataset $D_2$:

$$\hat{cdiv}(D_1, D_2) = \mathbb{E}_{B_1 \sim D_1, B_2 \sim D_2} d(\vec{f}_{B_1}, \vec{f}_{B_2})$$

We introduce these two definitions (diversity and cross diversity) to show our results hold with respect to two intuitive and logical ways to define data diversity (details in Appendix E). In addition, the *cross diversity* coefficient also measures the similarity/alignment or difference *between* datasets (using Task2Vec representations of data).

## 2.3 Backbone Used to compute the Diversity Coefficient

To compute Task2Vec embeddings, we use GPT-2 Radford et al. (2019) pre-trained on the English language as the probe network $f_w$. Following Task2Vec (Achille et al., 2019a), we fine-tune only the final layer (a language modeling head) on each batch since it's the only tested method for computing Task2Vec embeddings Achille et al. (2019a); Miranda et al. (2022a; 2023), e.g. it's not known if the intuitive properties observed in (Achille et al., 2019a) hold without fine-tuning the backbone. See Figure 1 for a visual of the diversity coefficient computation pipeline.

## 2.4 Recipe for Establishing if a Diversity Coefficient is High via the Conceptual Lower and Upper Bounds

To establish if a diversity coefficient $\hat{div}(D)$ of a dataset $D$ is high (or low), we use two conceptually well-motivated reference values. We call them the lower and upper bounds of the diversity

Table 1: **Diversity coefficients of LLM pre-training datasets are 2.7-4.76 times higher than the conceptual lower bound and more than half that of the upper bound. Cross diversity coefficients of LLM pre-training datasets are 3-5 times higher than the conceptual lower bound and more than half that of the upper bound. Overall, any strategy of concatenating datasets increases the cross diversity coefficient**. For the diversity coefficient, batches of text sequences were sampled from a mixed (i.e. interleaved) pool of sequences from all sub-datasets. Thus, sequences from both sub-datasets are present in the same batch at a rate dictated by the data mixture. Mix1 stands for a data mixture with ratio 3:1 (i.e., 0.75 to 0.25) for the corresponding combined data sets. Mix2 stands for a data mixture according to LLaMA v1 (i.e., 0.77, 0.23) for the corresponding combined data sets (see Appendix I.6 for details). Note, cross diversity does *not* mix datasets when computing the corresponding coefficient. Instead, we sample batches entirely from one of the sub-datasets; the distance between batches is then used to compute the cross diversity (see Appendix E for explanation).

| DATASET | DIVERSITY COEFF. | CROSS DIVERSITY COEFF. |
|---|---|---|
| LOWER BOUND (LB) | **0.0525** $\pm$ 3.41E-4 | (SAME AS LEFT) |
| NIH EXPORTER | 0.15 $\pm$ 3.218E-5 | (SAME AS LEFT) |
| USPTO | 0.1582 $\pm$ 4.09E-5 | (SAME AS LEFT) |
| PUBMED ABSTRACTS | 0.168 $\pm$ 2.63E-5 | (SAME AS LEFT) |
| HACKERNEWS | 0.201 $\pm$ 4.52E-5 | (SAME AS LEFT) |
| WIKITEXT-103 | 0.2140 $\pm$ 7.93E-5 | (SAME AS LEFT) |
| COMBINATION OF FIVE DATASETS (MIX2) | **0.217** $\pm$ 9.81E-4 | **0.2939** $\pm$ 2.03E-4 |
| SLIMPAJAMA | 0.221 $\pm$ 9.97E-4 | (SAME AS LEFT) |
| OPENWEBTEXT | 0.222 $\pm$ 1.00E-3 | (SAME AS LEFT) |
| C4 AND WIKITEXT-103 (MIX1) | **0.235** $\pm$ 1.04E-3 | **0.2711** $\pm$ 3.22E-4 |
| C4 | 0.2374 $\pm$ 2.785E-5 | (SAME AS LEFT) |
| THE PILE | 0.2463 $\pm$ 3.034E-5 | (SAME AS LEFT) |
| PILE-CC | **0.2497** $\pm$ 3.41E-5 | (SAME AS LEFT) |
| UPPER BOUND (UB) | **0.4037** $\pm$ 1.932E-5 | (SAME AS LEFT) |

coefficient. To understand the lower bound, consider a dataset constructed by sampling with most of the probability mass concentrated on some arbitrary token. This is a good candidate for a dataset with minimum diversity. To understand the upper bound, consider the other extreme: a dataset constructed by sampling any token uniformly at random given a fixed vocabulary (in our case, the GPT-2 tokenizer vocabulary) is a good candidate to create a dataset with maximum diversity.

Therefore, we measure a conceptual lower bound on a dataset with a vocabulary size of 2: <eos> token and a randomly selected non-special token from the GPT-2 tokenizer vocabulary. The <eos> token was assigned a probability weight of $1/\{\text{GPT-2 vocab size}\}$. The non-special token was assigned the remaining weight. Similarly, a high or maximum diversity dataset would consist of random sequences of all possible tokens, with no underlying order to semantics, formatting, etc. The upper bound of the diversity coefficient was therefore measured on a synthetic dataset with an equal probability of occurrence assigned to all tokens in the GPT-2 tokenizer vocabulary.

## 3 EXPERIMENTS & RESULTS

### 3.1 PRE-TRAINING IN HIGHER DIVERSITY LEADS TO BETTER EVALUATION PERFORMANCE

Next, we highlight one of our main empirical results, showing that pre-training from scratch on data sets of **increasing formal diversity** leads to **better evaluation performance** on the validation set. We want to emphasize that our results are *interventional and not correlative*— i.e., we intervene by constructing pre-training data sets of increasing diversity, train the models from scratch on these (controlling tokens), and finally demonstrate performance improves.

**Experiments:** We computed the formal diversity coefficient (described in section 2.2) of three publicly available datasets (USPTO, PubMed, and the combination of USPTO+PubMed) with diversity values of 0.158, 0.168, and 0.195 respectively. We then pre-trained GPT2s (Radford et al., 2019) models of size 51M, 117M, 204M, 345M, 810M, 1.5B parameters and LLaMAv2 7B (Touvron et al.,

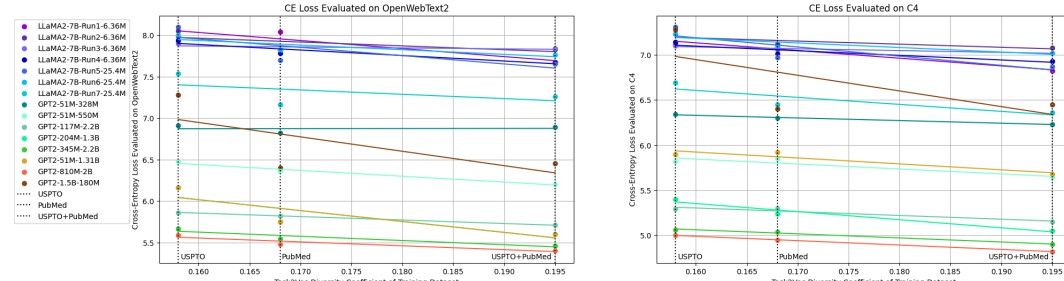

Figure 2: **Pre-training on datasets with higher diversity coefficients leads to better evaluation performance on diverse datasets.** Legends are formatted "Model Type-Number of Parameters-Number of Tokens trained on". The models were evaluated on the validation sets. The diversity of the pre-training datasets increase from $0.158$, to $0.168$, to $0.195$. Models are evaluated on the validation split of the given evaluation dataset (given in each plot's title). All models of the same run (i.e. same color) are trained on the same number of tokens and the same hyperparameters, minimizing cofounding factors.

2023b) from scratch. To ensure that the diversity of the dataset is the primary factor driving our results, we train all models using a fixed number of tokens and the same hyperparameters across the same experiment. An experiment is considered to be the triplet of models trained with the USPTO, PubMed, and USPTO+PubMed. The 51M model is trained on 550M tokens, the 117M model is trained on 2.2B tokens, the 204M model is trained on 1.3B tokens, the 345M model is trained on 2.2B tokens, the 810M model is trained on 2B tokens, and the 1.5B model is trained on 180M tokens. The LaMMAv2 7B model is trained on 6.36M tokens. For training runs with combined datasets, each batch has interleaved examples from both datasets with equal mixing proportions.

**Results:** Figure 2 demonstrates that as the diversity coefficient of the pre-training dataset increases, the evaluation performance improves. This key result follows because the cross-entropy loss on the validation falls on all 15 runs in 2. We evaluated on the validation set of C4 and OpenWebText2 which have diversity coefficients of 0.237 and 0.222. We deliberately avoided selecting the Pile in order to minimize contamination issues that could potentially confound our results. We also deliberately selected dataset with high diversity because the conjecture is that diverse pre-training datasets help the most on diverse general evaluation benchmarks. Our results also have positive average $R^2$ values of approximately 0.79 for OpenWebText2, 0.82 for C4 when including all 8 GPT-2 experiments—all trained for over 150M tokens, and most over 1B tokens—and similar $R^2$ values when also including all remaining LLaMA2 experiments (larger models trained on fewer tokens).

### 3.2 DIVERSITY COEFFICIENTS OF PRE-TRAINING DATA SHOWS LLMS ARE PRE-TRAINED ON FORMALLY HIGHLY DIVERSE DATA

**Experiments:** We evaluate the diversity coefficient and cross diversity coefficient (both described in section 2.2) of ten publicly available LLM pre-training datasets (described in section D). We also compute the diversity and cross diversity coefficients of two concatenated datasets: 1) C4 and WikiText-103, and 2) five sub-datasets of The Pile: Pile-CC, HackerNews, NIH ExPorter, PubMed, and USPTO (Appendix I.4). In addition, we compute our conceptually well-motivated lower and upper bounds on the diversity coefficient (section 2.4).

**Results:** Table 1 reports the aforementioned diversity and cross diversity coefficients. The key observations from our results are:

- The cross diversity coefficients of pre-training datasets tend to be **3-5 times greater than the theoretical lower bound and, on average, half the upper bound.** Prominently, C4, The Pile, and Pile-CC exhibit the highest diversity coefficients (0.23 - 0.25). This aligns with intuition, as these are very large, web-crawl-based, internet-scale datasets.

- The diversity coefficients of pre-training datasets tend to be **2.7-4.76 times greater than the theoretical lower bound and, on average, half the upper bound.** As expected, this

is slightly lower than the cross diversity coefficient because the cross diversity coefficient compares batch embeddings from different, disjoint datasets.

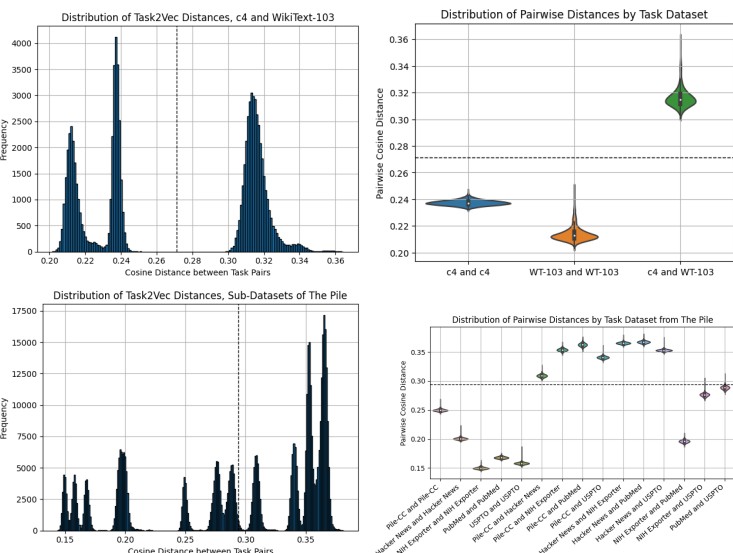

Figure 3: **Distribution of pairwise batch distances reflects conceptual and semantic dataset properties, therefore increasing trust in the diversity and cross diversity coefficient.** Pairwise task distances from the concatenated C4 and WikiText-103 dataset (top) and concatenated five sub-datasets of The Pile (bottom) take on a multi-modal form according to dataset comparisons. Pairwise distances are segmented by source datasets for each pair of batches (right), where each sub-distribution corresponds to a mode from the histograms (left). Dotted lines denote the cross diversity coefficient of the concatenated C4 and WikiText-103 dataset (top) and concatenation of five sub-datasets of The Pile (bottom). These results show that combining batches from two different datasets computes a higher cross diversity, as expected. Therefore, these results align with human intuition, increasing the confidence in the diversity and cross diversity coefficients as diversity metrics.

### 3.3 CONCATENATION OF DATASETS OF DIFFERENT SOURCES PRODUCES HIGHER MEASURED DIVERSITY

**Experiments:** To show that the concatenation of different datasets produces high diversity datasets, we measure the cross diversity coefficient of C4 plus WikiText-103, as well as of the five sub-datasets of The Pile in Table 1. To understand the source of this increased diversity, we plot in Figure 3 the Task2Vec (cosine) distances between batches from individual datasets and distances of batches from the different datasets.

**Results:** Our key observations are:

- The cross diversity coefficient for the C4 and WikiText-103 concatenated dataset is 0.2711, about +0.03-0.05 higher than that of each individual dataset.

- The cross diversity coefficient for the concatenation of the five sub-datasets of the Pile is 0.2939 (Table 1), which is about +0.04-0.1 (Figure 3) that of each individual dataset.

This increase in diversity occurs because concatenating datasets produces higher pairwise Task2Vec distances between batches from different datasets (see Figure 3). Note that, this aligns with human intuition that combining data from heterogeneous sources increases the overall diversity of the data.

## 3.4 DISTRIBUTION OF PAIRWISE BATCH DISTANCES REFLECTS CONCEPTUAL AND SEMANTIC DATASET INFORMATION

To increase our confidence in the diversity and cross diversity coefficient as diversity metrics, we study distributions of the Task2Vec (cosine) distances used to compute the coefficient. We examine the alignment of the grouping of these distances with (human) conceptual and semantic understanding in Figure 3.

**Experiments:** We analyze Task2Vec (cosine) distances between batches from five sub-datasets of The Pile. We compare distances between batches of individual sub-datasets and distances across different sub-datasets.

**Results:** Our key observations are:

- Figure 3 (top, left) shows 3 modes. We confirm that the modes correspond to pairings of datasets in Figure 3 (top, right). For instance, the right-most mode, corresponding to distances with values higher than the cross diversity coefficient, consists of pairwise distances between C4 and WikiText-103 batches. This confirms intuitive properties we'd expect, i.e. we'd expect 3 modes given 2 datasets ($C_2^2 + 2 = 3$).
- Similarly to the preceding point, Figure 3 (bottom, left) shows 15 modes, which is exactly the number expected in enumerating all possible pairs of batches from 5 datasets.[1] Due to overlaps in distance values we only see 11 modes in the Figure 3 (bottom, right).

These findings build trust in the cross diversity coefficient as a dataset diversity metric, since the coefficient and underlying Task2Vec distances of batches behave in interpretable ways that align with human intuition. Since the diversity coefficient uses the same computational backbone as cross diversity, these findings also build trust in the diversity coefficient.

## 3.5 DIVERSITY COEFFICIENT CAPTURES LLM PRE-TRAINING DATA DISTRIBUTIONAL PROPERTIES

To instill further confidence in the diversity coefficient, we perform a correlation analysis with data distributional properties on the GINC dataset synthetic language dataset Xie et al. (2021). GINC generates sequences by modeling how real documents are generated given a fixed number of latent document concepts through a mixture of Hidden Markov Models (HMM), where each HHM has a latent concept that models document statistics, e.g. wiki bio. Further details on GINC can be found in section K.

**Experiments:** Given that each GINC dataset is a mixture of HMMs with a fixed number of latent concepts (1-10,000), we plot how the diversity coefficient varies as the number of latent concepts increases for each dataset. We plot this in Figure 4 (top) and fit a curve for GINC datasets with fixed vocabulary sizes of 50 and 150. Then we fix the number of latent concepts at 5 and 5000 and similarly plot how increasing the vocabulary size for the GINC dataset (50-10,000 unique tokens) increases the diversity coefficient. We plot this in Figure 4 (bottom) and fit a curve for GINC datasets with 5 latent concepts and 5000 latent concepts.

**Results:** Our observations are as follows:

- **Diversity coefficient increases with greater number of latent concepts.** Figure 4 (top) shows adding more latent concepts increases the diversity coefficient with diminishing returns. We hypothesize that additional latent concepts introduce new and varied document-level statistics, resulting in an increase in the diversity coefficient. The $R^2$ is high, with values 0.952 and 0.898.
- The diversity coefficient eventually saturates as more latent concepts are added. We hypothesize this is due to the small size of a synthetic data set vs. a real one.
- **Diversity coefficient increases with larger vocabularies.** Figure 4 (bottom) shows the measured diversity coefficient increases at a seemingly exponential pace for larger vocab sizes. The $R^2$ is high with values 0.993 and 0.984.

---

[1]Given a 5 by 5 distance matrix, we'd expect the lower triangular portion plus the diagonal to be the number of pairings, so $C_2^5 + 5 = 15$.

These results show the diversity coefficient successfully captures different distributional sources of variation of the data.

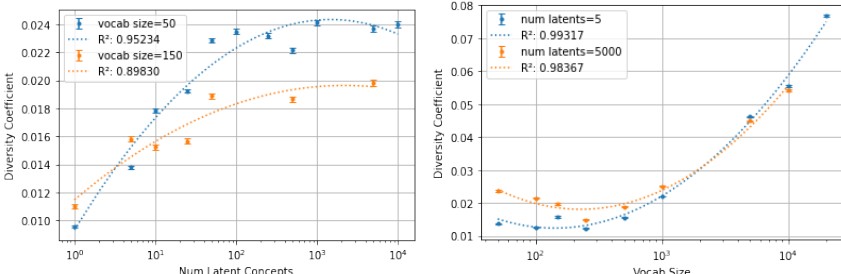

Figure 4: **Diversity coefficient of GINC datasets with varying number of latent concepts and vocab sizes shows the diversity coefficient behaves as expected.** The diversity coefficient increases and saturates with an increasing number of latent concepts (top) and exponentially increases with increasing vocab size (bottom). This implies that increases in the measured diversity coefficient correspond to changes in LM pre-training data distributional properties that intuitively enable more diverse data.

## 4 RELATED WORK

Existing diversity metrics have concentrated on data produced by Generative Adversarial Networks (GANs) and involve variations of a precision- and recall-based framework originally proposed in (Sajjadi et al., 2018) to measure quality and diversity, respectively (Kynkäänniemi et al., 2019; Simon et al., 2019; Naeem et al., 2020). Similar to our metric, these methods use embedding functions. These methods argue data quality is not synonymous with data diversity in the context of GANs (Fowl et al., 2020) and take a two-metric approach. Regarding LLMs, we argue that data diversity is a subset of data quality, which is demonstrably important to enable emergent capabilities such as in-context learning Xie et al. (2022); Shin et al. (2022); Chan et al. (2022). Recent work has also confirmed the importance of diversity from the perspective of deduplication (Tirumala et al., 2023), and the general importance of quantitatively informed data selection (Xie et al., 2023a). Hence, diversity metrics capture an important aspect of data quality.

A recently proposed diversity metric that doesn't rely on an embedding function is the Vendi Score (Friedman & Dieng, 2022), which is the exponential of the Shannon entropy of the eigenvalues of a similarity matrix/kernel. For more discussion of related work, please see Appendix B.

## 5 DISCUSSION

Our work extends, examines, and thus validates the application of the Task2Vec diversity coefficient to a new modality—natural language—and demonstrates that open LLMs are pre-trained on formally diverse data.

One potential limitation of our method is the need for a data representation. Although the requirement for a data representation might seem restrictive, we argue that it is an inherent aspect of data processing. Choosing symbols (e.g., one-hot vectors), raw pixels, etc. **is** a choice of data representation. We suggest deep learning representations due to their overwhelming success in machine learning, e.g. in computer vision (Krizhevsky et al., 2012; He et al., 2015), natural language processing (Devlin et al., 2018; Brown et al., 2020; Chowdhery et al., 2022; OpenAI, 2023; Google, 2023), game playing (Silver et al., 2016; Mnih et al., 2013; Ye et al., 2021), theorem proving (Rabe et al.; Polu & Sutskever, 2020; Han et al.), code (Chen et al.) and more. In addition, widely available open-source pre-trained models (e.g. CLIP (Radford et al., 2021), LLaMA (Touvron et al., 2023a), etc.) have made choosing a good embedding method easier. Another potential limitation of the diversity coefficient is the need to fine-tune a probe-network. Though this does introduce some computational overhead, it is relatively small (fine-tuning only the final layer) and outweighed by the utility of using information-rich Task2Vec embeddings.

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

## A  INTUITION OF TASK2VEC

To better understand the Task2Vec embedding, observe that the (diagonal) of the FIM can be interpreted as a measure of the information that a given parameter contains about the generative distribution $p_w(\hat{x}_t \mid x_{t-1:1})$. Therefore, it serves as a unique fingerprint, or feature vector, for a batch, which defines a task distribution; as such, this vector, in its entirety, is as large as the number of parameters of the probe network. Empirical findings in (Achille et al., 2019a) show that Task2Vec embeddings cluster in a way that reflects semantics between different visual concepts, and that Task2Vec cosine distances are positively correlated with taxonomical distances.

We will provide a rephrasing from the original Task2Vec Achille et al. (2019a) explaining why Task2Vec is captures the importance of the weights and therefore why it serves as a unique fingerprint for a task. The importance of a weight in a network can be measured by how much the predictions changes (average KL divergence of the original vs perturbed output) with respect to the loss of interest to target task. Consider a perturbation of the weights $w' = w + \delta_w$. This is the a 2nd-order approximation to the change in outputs:

$$E_{x \sim \hat{p}}[KL(p'_w || p_w)] = \delta_w F \delta_w^\top + o(\delta(w)^2)$$

Therefore, the information content of the weights is measured by this change in output captured by the Fisher Information Matrix. Therefore, this serves as a unique finger-print of which parameters are important, making the diagonal of F a strong candidate for the unique fingerprint of a task via importance of weight. For a relation between Task2Vec and Kolmogorov Complexity, see section C.3.

## B  RELATED WORK (CONT.)

Building on the success of deep learning data representations, we demonstrate deep learning is a strong way to create dataset/task embeddings. In contrast to the Vendi Score, our approach learns effective embeddings of datasets in an end-to-end manner, whereas the Vendi Score is focused on measuring diversity between specific data points. Since many datasets are publicly available (e.g. Common Crawl, Wikipedia), data used to train new models may be curated from such datasets, necessitating a metric that captures overall data diversity. These scenarios are thus in favor of using the Task2Vec diversity coefficient. Therefore, our method is likely more general and scalable than the Vendi Score. We leave a detailed comparison with the Vendi Score as future work.

Returning to the utility of a practical diversity metric, there is strong empirical support from efforts at increasing multi-modal image and text data quality that using *semantically* deduplicated data (as opposed to simply exact *match* deduplication) for training can yield impressive training efficiency gains and even improve out of domain generalization (Abbas et al., 2023; 2024). Though motivated by the same underlying hypothesis about the importance of diversity in data quality, our work on the diversity coefficient focuses on text-only data, the standard for training LLMs, and yields a more interpretable metric that allows practitioners to rigorously *understand* the quality of datasets, enabling the informed *choice* of baseline higher quality datasets rather than only being able to *improve* a *chosen* dataset. Furthermore, the generality of the diversity coefficient allows it to assess and inform the results of traditional data *pruning*, which has been shown to be a promising method for increasing data quality (Sorscher et al., 2023), while also lending itself naturally as a useful metric for guiding diverse data corpora *construction*, which has also been seen to be a fruitful endeavor with the release large, diverse, open source corpora (Gao et al., 2020).

More generally, better understanding and clarity about the characteristics of datasets and data quality enable greater control and direction of model capabilities (Mitchell et al., 2023), which become especially important as models become more complex, more powerful, and more important to scientific and economic activity.

However, the benefits of this more sophisticated aggregation method are not clear, and its computation ($O(n^3)$) is more expensive than the diversity coefficient ($O(n^2)$). Moreover, it assumes a suitable similarity function/kernel, and does not provide guidance on data representation, arguably the most important ingredient in machine learning. Furthermore, they suggest that utilizing data representational methods such as embedding networks that require pretrained models may be limiting. We argue instead that data representation is a fundamental property of data processing that has

led to the overwhelming success in machine learning due to deep learning, e.g. in computer vision (Krizhevsky et al., 2012; He et al., 2015), natural language processing (Devlin et al., 2018; Brown et al., 2020; Chowdhery et al., 2022; OpenAI, 2023; Google, 2023), game playing (Silver et al., 2016; Mnih et al., 2013; Ye et al., 2021), theorem proving (Rabe et al.; Polu & Sutskever, 2020; Han et al.), code (Chen et al.) and more. For further comments on the Vendi Score see section B.

# C  Justification of Task Based (Task2Vec) Diversity vs Activation, Token, and Model Agnostic based Diversity Formalizations

One could potentially have used different data and task representations to formalize diversity. In this section, we explain the reasons for using Task2Vec compared to other reasonable representations. We explain it compared to 3 potential alternatives: 1. Activations as data representation 2. Diversity based on token distribution 3. Model Agnostic Distributions, e.g., using Kolmogorov Complexity/minimum description lengths.

## C.1  Resolution Power Deficits in Activation-Based Diversity Approaches Compared with Task2Vec Diversity

**Result:** We discover that activation-based diversity lacks the resolution to detect changes in diversity at a granular level. We demonstrate this in figure 5 where the activation-based diversities become flat quickly, while Task2Vec diversity (the diversity coefficient) varies more smoothly and detects diversity changes for longer.

**Method:** The alternative definition of diversity we explored was computed by taking the average distance between a pair of batches of sequences of text and using GPT-2's final layer as the data representation of the batch:

$$\hat{d}\text{iv}(D) = \mathbb{E}_{B_1, B_2 \sim D} d_{metric}(GPT2^{(L)}(B_1), GPT2^{(L)}(B_2))$$

Where $B_1, B_2$ is a standard batch of text sequences, $D$ is the data distribution, $metric$ is Linear Centered Kernel Alignment (CKA) (Kornblith et al., 2019) or PW Canonical Correlation Analysis (PWCCA) (Morcos et al., 2018), $GPT2^{(L)}(B1)$ is the activation matrix of the final layer $L$ after reshaping it to size $R^{BT \times D}$ where $B$ is the batch-size, $T$ is the sequence length and $D$ is the size the final layer activations (768 for GPT2).

**Experimental Setup:** We varied the size of the vocabulary size $|V|$ from $2/|V|$ to $0.4|V|$ using 100 evenly spaced samples. We focused only on that range because CKA and PWCCA became flat very quickly—just after $8.09\%$ (at 0.0323), while the Task2Vec diversity did not become flat but instead followed close to a linear trend. The data generation distribution $D$ was uniform independently sampled random tokens. The batch-size $B$ was 32 with sequence length $T$ equal to 240 and the number of batches we used for the expectation was 30. We also made sure the dimensionality $D = 768$ of the activation matrix was 10 times smaller than the $TB = 240 * 32 = 7,680$. This last condition is crucial for CKA, PWCCA distances to not be vacuous Raghu et al. (2017); Miranda et al. (2022a). Intuitively, this happens because one has more features than effective data points and say CCA tries to maximize linear correlation vectors between the two datasets that can always be done perfectly with too many features.

**Experimental Interpretation:** We hypothesize that Task2Vec has more resolution than activation-based diversities because Task2Vec attempts to estimate an approximately unique fingerprint of the generating parameters of the batch. However, activation instead has been optimized for next token prediction/discrimination (or in classical supervised learning for classification), i.e., features are obtained that discriminate well between data classes/vocabulary indices. Therefore, one would expect consistently high distances between activation-based features—which is what we approximately observe, after only $8.09\%$ of the x-axis, the activations based diversity (average activation distance) achieves the maximum value and remains constant.

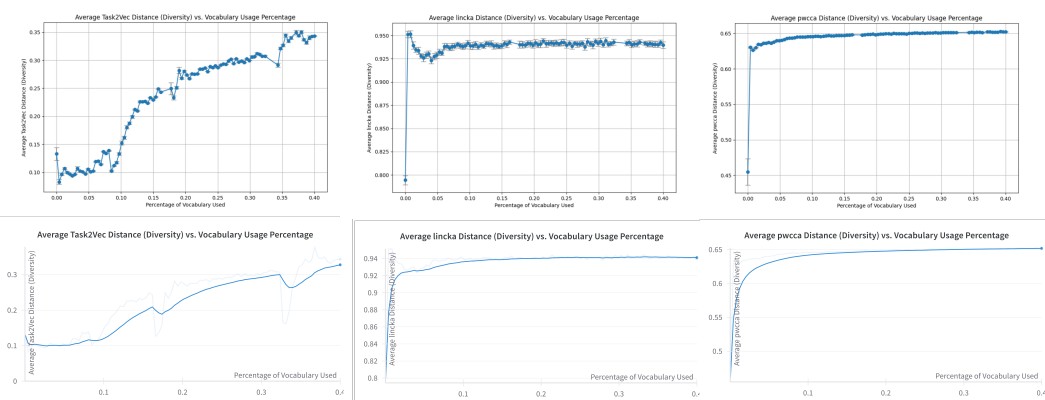

Figure 5: **Shows the resolution power deficit in activation-based diversity approaches compared to Task2Vec Diversity.** We plot the diversity on the y-axis against percentage of the vocabulary used on the x-axis. Varying the vocabulary varies the diversity of a data set because the smaller the vocabulary, the more homogenous the sequence of text will be. For example, vocabulary size 1 gives the same sequence while $|V|$ allows for $|V|^T$ possible sequences. After only $8.09\%$ of the x-axis, the activation based diversity increased to it's maximum values and remains approximately constant within 95%-confidence intervals, while Task2Vec diversity increases smoothly. To make the patterns clearer, we smoothed out the top three plots by removing outliers and used a smoothing value of 0.79 in the bottom three plots. The outlier plots can be recovered by observing at the light blue line in the bottom three plots. We conjecture that the outlier were caused by the small number of batches we used (30).

## C.2 Token Based Definitions of Diversity are Brittle to Paraphrasings

One can also consider formalizations of diversity based on token distributions in a batch (or data set). We did not use this because previous work has already demonstrated its limitation as data representations for train-test data contamination detection (Jiang et al., 2024). This is highly related to our work because contamination detection requires good data representations for detecting differences in text. In particular, they say: "these contamination definitions primarily revolve around n-gram or token overlaps, which only target direct duplications present in both training and evaluation datasets and might provide both high false positive rate (since many semantically different texts have overlaps) and false negative rate (since simple paraphrasing can evade detection)". To make it more concrete, consider an example. Consider the sentences "The great paper" and "the fantastic paper". These would be binned to entirely different locations in a histogram, while the difference in the sentence is varying much more smoothly. Therefore, token based representation of data are brittle to simple paraphrasing and as our example shows result in overly "harsh" metrics—similar to previous work that criticizes harsh metrics for LLM evaluation (Schaeffer et al., 2023a). In addition, previous work has shown that importance weights of data like text are statistically intractable to estimate without sufficient additional structure (Xie et al., 2023b; Bengtsson et al., 2008; Gelman & Meng, 2004; Snyder et al., 2008). For example, the histograms of different types of text end up being sparse, even with hashing e.g., consider 50k (GPT2), 30K (LLaMA2) or 10K (DSIR) feature vectors. We conjecture that if these representations were robust, the transformer architecture (Vaswani et al., 2017) wouldn't have emerged as the successful architecture that it is today (2024).

## C.3 Discussion on Model Agnostics Formalizations of Data Diversity

**Theoretical Relation of FIM (Model) and Kolmogorov (Model Agnostic) based metric:** Theory can be important to have a better understanding of the world. Therefore, theoreticians might be interested in diversity definitions that are model agnostic, since, they might be interested in the inherit diversity of the data or task. We emphasize that Task2Vec was derived with such considerations in mind: "The FIM is also related to the (Kolmogorov) complexity of a task, a property that can be used to define a computable metric of the learning distance between tasks" (Achille et al., 2019a). More precisely (Achille et al., 2018) showed that the training dynamics of a deep network minimiz-

ing a loss function can be viewed as approximately optimizing an upper bound on the Kolmogorov Structure Function (KSF) of the learning task. This connection arises because the loss function incorporates a complexity term measured by the Fisher Information Matrix of the network weights. Therefore, Fisher information provides a computable way to relate the geometry of the loss landscape to notions of task complexity linked to Kolmogorov complexity. Therefore, Task2Vec (via FIM) was derived with Kolmogorov Complexity in mind. In Summary, the relationship between Fisher Information Matrix (FIM) and Kolmogorov Complexity, as discussed in Achille's works (Achille et al., 2019a; 2018; 2019b; 2021), suggests that FIM can serve as an upper bound for Kolmogorov Complexity in the context of learning tasks. Specifically, the FIM captures the sensitivity of the learning model to its parameters, reflecting the amount of information about the data that the model encodes. This sensitivity can indirectly bound the Kolmogorov Complexity by quantifying the model's complexity and its capacity to learn from data, thus offering a bridge between empirical measures of complexity (FIM) and theoretical ones (Kolmogorov Complexity). However, explicit relationships quantifying how distances based on Fisher information matrices correlate with true Kolmogorov complexity are not provided.

**Remarks on approximations to Kolmogorov:** The central way we control for our results to be model agnostic in practice is by fixing the model. This means that the data is the only thing being varied, and the features being used for any diversity of computations are consistent. Since, Kolmogorov Complexity in uncomputable we conjecture this is approximately optimal—given that it's essential in practice to have a way to represent the data.

# D    LLM Pre-training Datasets

Since LLMs are often trained on internal, non-public datasets[2], we used publicly available language datasets from the same sources as LLM pre-training data:

**C4**, a 305GB cleaned version of Common Crawl's web crawl corpus in English Raffel et al. (2019). Sequences in C4 were extracted from the web via de-duplication methods and heuristics to remove boiler-plate and gibberish.

**WikiText-103**, a 500MB collection of over 100 million tokens extracted from the set of verified Good and Featured articles on Wikipedia Merity et al. (2016).

**The Pile**, a 825 GiB open-source English-text corpus for language modeling that combines 22 smaller, high-quality datasets from diverse sources Gao et al. (2020). These sources include Pile-CC (Common Crawl), PubMed Abstracts, Books3, OpenWebText2, ArXiv, and GitHub.

**SlimPajama**, an open-source reproduction of the LLaMA v1 dataset with extensively deduplication. It incorporates data from CommonCrawl (CC), C4, GitHub, Books, ArXiv, Wikipedia, and StackExchange, totaling 627 billion tokens Soboleva et al. (2023).

For instance, GPT-3 was trained on a filtered Common Crawl dataset and Wikipedia Brown et al. (2020), which are represented by C4 and WikiText-103. It was also trained on WebText2 and Books, which are sub-datasets of The Pile.

We also evaluate the diversity coefficient of the following six sub-datasets of The Pile:

**Pile-CC**, a 227 GiB preprocessed version of Common Crawl's web crawl corpus Gao et al. (2020). While both Pile-CC and C4 are sourced from Common Crawl, Pile-CC was preprocessed from Web Archive files, which are raw HTTP responses and page HTML, whereas C4 was preprocessed from WET files, which consist of plaintext. Nonetheless, we expect that both datasets are non-mutually-exclusive.

**HackerNews**, a 4 GiB scraped and parsed dataset of comment trees from Hacker News, a social news website that aggregates article links Gao et al. (2020). Articles are generally focused on topics in computer science and entrepreneurship.

**NIH ExPorter**, a 1.9 GiB dataset of NIH Grant abstracts for awarded applications from 1985-present hosted on the ExPORTER initiative Gao et al. (2020).

---

[2]For instance, Gopher was trained on Google's internal dataset MassiveText.

**PubMed Abstracts**, a 19 GiB dataset of abstracts from 30 million publications in PubMed Gao et al. (2020).

**USPTO Backgrounds**, a 23 GiB dataset of background sections from patents granted by the United States Patent and Trademark Office (USPTO) Gao et al. (2020).

**OpenWebText2**, a 38 GiB dataset based on data extracted from Reddit posts, deduplicated, and filtered for English content using FastText. Strict filtering with local-sensitive hashing (LSH) was done and only unique content with similarity of less than 0.5 was used. The finalized dataset comprises 38 GiB.

## E    FURTHER JUSTIFICATION AND EXPLANATION OF CROSS DIVERSITY

We include the notion of the cross diversity coefficient as it also leverages the ability of distance between Task2Vec embeddings to capture important properties of the data, and allows one to more clearly assess similarity/diversity *between* datasets (hence the term *cross* diversity), as opposed similarity/diversity *within* datasets (the focus of the "normal" diversity coefficient). Thus, when information on clearly defined sub-datasets is available, one can use cross-diversity to more specifically determine the diversity of the concatenation of the given datasets (to form the total/overall dataset). Thus, the cross diversity coefficient provides us the opportunity to more robustly characterize methods of defining data diversity by offering a different perspective on combined datasets from the diversity coefficient. Note that in the case $D_1 = D_2$, i.e. one is calculating the cross diversity of a single dataset (with itself), cross diversity becomes equivalent to diversity.

## F    TASK2VEC DIVERSITY COEFFICIENT CORRELATES WITH GROUND TRUTH DIVERSITY

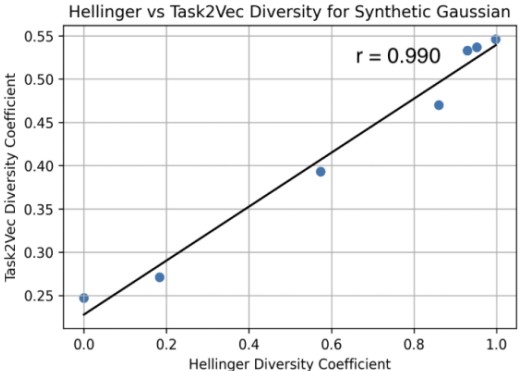

Figure 6: **Task2Vec diversity coefficient correlates with ground truth diversity for synthetic Gaussian benchmark.** Source: Miranda et al. (2022b)

As shown in Miranda et al. (2022b), when the ground truth diversity is available for a synthetic Gaussian benchmark, the Task2Vec diversity coefficient correlates with the ground truth diversity. These results provide confidence in the Task2Vec diversity coefficient as diversity metric.

Figure 1 shows our pipeline for computing the diversity coefficient of large scale, natural language datasets. See section 2.2 for more details on our method.

## G    HIGHER FORMAL DIVERSITY OF THE TRAINING DATA SET CAUSES HIGHER TEST PERFORMANCE

Table 2 demonstrates the causal relationship between pre-training data diversity and downstream evaluation performance on a diverse target domain.

| USPTO Evaluation Example | Wikitext-103 Evaluation Example |
|---|---|
| **Context**:
1. Field of the Invention\nIn an extensive plant breeding program, Grant Merrill, originator and now deceased, originated a large number of new and distinct varieties of fruit trees, and which included the herein-claimed variety of peach tree. Such plant breeding program was undertaken in originator's experimental orchard | **Context**:
Traditional Chinese literary criticism emphasized the life of the author when interpreting a work , a practice which Burton Watson attributes to " the close links that traditional Chinese thought posits between art and morality " . Since many of Du Fu 's poems feature morality and history , this practice is particularly important . Another |
| **Pubmed-trained model output**:
, and the results of the present study suggest that | **Pubmed-trained model output**:
aspect of the author's work is that the author |
| **USPTO-trained model output**:
plant breeding program.\n2. Description of the | **USPTO-trained model output**:
problem with the present invention is that the present invention |
| **Pubmed & USPTO interleaved-trained model output**:
breeding program.\n2. Description of the Related | **Pubmed & USPTO interleaved-trained model output**:
issue is the fact that the author is not familiar |
| **Ground truth text**:
located near Exeter, Tulare County, Calif.\n2. Prior | **Ground truth text**:
reason , identified by the Chinese historian William Hung , is |

Figure 7: **Despite smaller scale (51M parameters) and relatively high loss/perplexity on certain evaluation datasets, the experimental models still grasp important aspects of sentence syntax and structure, even on evaluation data they otherwise struggle on (e.g. Wikitext-103, for which all models scored around 6.6 loss or more)**. In addition, when evaluated on similar, well aligned datasets (e.g. USPTO + PubMed Abs. (train) on USPTO (validation)), models are sometimes also able to match the form and semantic content of the example. Input examples were randomly picked from the given evaluation dataset.

## G.1 Higher Formal Diversity of the Training Data Set Causes Higher Test Performance: GPT2 Results

Table 2: **The table illustrates that as the diversity coefficient of training data increases, the cross-entropy loss (CE) decreases on diverse tasks, implying that training on higher diversity data improves evaluation performance on diverse tasks.** Each model is trained for the same number of tokens (1.31 B tokens total), all models have 51.5 M parameters, and all use a GPT-2-based architecture; all other hyperparameters are identical and controlled for. We evaluate on Pile-CC (Pile Common Crawl) and OpenWebText2 since those datasets align with intuitively diverse datasets, which we verify with the computed diversity values shown.

| Training Data Set | OpenWebText2 (**div 0.222**) |
|---|---|
| USPTO (**div 0.158**) | 6.4414 |
| PubMed (**div 0.168**) | 6.4204 |
| USPTO + PubMed (**div 0.195**) | **6.1815** |

## G.2 Higher Formal Diversity of the Training Data Set Causes Higher Test Performance: All Results

Table 3 demonstrates how the cross-entropy of the validation loss decreases as the diversity of the pre-training data set increases. All tokens were fixed. The table includes LLaMAv2 7B results trained from scratch.

Table 3: **Cross-entropy of the validation loss decreases as the diversity of the pre-training data set increases**. We report the test cross-entropy (no parenthesis) and the corresponding perplexity (in parentheses) on the validation set of C4, OpenWebText2). The diversity of the evaluation data sets, C4 and OpenWebText2 is 0.222, 0.237 respectively. The pre-train diversity of USPTO, PubMed, USPTO+PubMed is 0.158, 0.168, and 0.195 respectively. All evaluations were on the full validation set except for the ones denoted as [8B] which used a batch size of size 8 due to runs crashing due to random hardware failure when running such large evaluations. C4's validation set has 365K examples. OpenWebText2 had 33.4k examples. We explicitly avoided evaluating on the pile to minimize cofounders on our experiments, e.g., that the good performance might be due to data contamination. We also evaluated on two data sets for the same reasons and to increase robustness of our observations. Table entries are continued in Table 4.

| Architecture | # Tokens (Seq Len) | Train Dataset | OWT2 (**div 0.222**) | C4 (**0.237**) |
|---|---|---|---|---|
| LLaMA2-7B (Scratch, run1) | 6.36M (1024 L) | USPTO | 7.99 (2959.1) | 7.22 (1372.9) |
| - | - | PubMed | 8.038 (3097.6) | 7.26 (1424.3) |
| - | - | USPTO + PubMed | 7.67 (2144.2) | 6.90 (999.6) |
| LLaMA2-7B (Scratch, run2) | 6.36M (1024 L) | USPTO | 8.048 (3127.0) | 7.23 (1385.5) |
| - | - | PubMed | 7.82 (2496.0) | 7.11 (1225.0) |
| - | - | USPTO + PubMed | 7.83 (2154.6) | 7.08 (1189.5) |
| LLaMA2-7B (Scratch, run3) | 6.36M (1024 L) | USPTO | 7.90 (2706.2) | 7.10 (1216.5) |
| - | - | PubMed | 7.82 (2513.4) | 7.06 (1172.8) |
| - | - | USPTO + PubMed | 7.84 (2548.8) | 7.02 (1126.7) |
| LLaMA2-7B (Scratch, run4) | 6.36M (1024 L) | USPTO | 7.94 (2826.9) | 7.14 (1267.2) |
| - | - | PubMed | 7.78 (2396.3) | 7.01 (1112.6) |
| - | - | USPTO + PubMed | 7.67 (2154.6) | 6.93 (1030.4) |

# H USING THE DIVERSITY COEFFICIENT IN PRACTICE: SETTING BATCH SIZE AND NETWORK PARAMETERS

**Experiments:** We test the sensitivity of the computed diversity coefficient value to changes in batch size and probe network parameters in order to gauge how these parameters should be set in practice for natural language datasets.

We vary the batch size and observe the impact on the diversity coefficient. For the same number of batches (200) and probe network (pretrained, fine-tuned GPT-2), we computed the diversity coefficient of C4 for batch sizes of 128, 256, 512, and 1024, and plot the results in Figure 8 (left).

We test the following probe network configurations to measure the diversity coefficient of C4 and of WikiText-103: 1. Pretrained GPT-2 with fine-tuning, 2. Pretrained GPT-2 without fine-tuning, 3. Randomly initialized GPT-2 with fine-tuning, 4. Randomly initialized GPT-2 without fine-tuning. Since using a random and/or non fine-tuned network is more resource efficient and easily accessible in practice, our motivation is to assess the necessity of using pre-trained and fine-tuned probe network, which is the original configuration used for Task2Vec in Achille et al. (2019a). We aim to determine if a good approximation of diversity can be computed without fine-tuning. This setting is shown in Figure 8 (right).

**Results:** We observe that

Table 4: **Continuation of Table 3.**

| Architecture | # Tokens (Seq Len) | Train Dataset | | OWT2 (**div 0.222**) | C4 (**0.237**) |
|---|---|---|---|---|---|
| LLaMA2-7B (Scratch, run5) | 25.4M (1024 L) | USPTO | | 8.10 (3302.1) | 7.31 (1497.0) |
| - | - | PubMed | | 7.70 (2217.1) | 6.97 (1067.7) |
| - | - | USPTO | + | 7.65 (2108.8) | 6.87 (966.8) |
| | | PubMed | | | |
| LLaMA2-7B (Scratch, run6) | 25.4M (1024 L) | USPTO | | 8.00 (2999.6) | 7.23 (1385.0) |
| - | - | PubMed | | 7.82 (2511.3) | 7.09 (1205.2) |
| - | - | USPTO | + | 7.76 (2351.7) | 7.02 (1120.1) |
| | | PubMed | | | |
| LLaMA2-7B (Scratch, run7) | 25.4M (4096 L) | USPTO | | 7.54 (11881.9) | 6.69 (809.5) |
| - | - | PubMed | | 7.16 (1293.7) | 6.45 (634.1) |
| - | - | USPTO | + | 7.26 (1425.0) | 6.36 (578.2) |
| | | PubMed | | | |
| GPT2-51M | 328M (1024L) | USPTO | | 6.91 (998.6) | 6.34 (568.3) |
| - | - | PubMed | | 6.82 (912.5) | 6.30 (543.2) |
| - | - | USPTO | + | 6.89 (984.3) | 6.23 (505.7) |
| | | PubMed | | | |
| GPT2-51M | 550M (1024L) | USPTO | | 6.47 (646.6) | 5.82 (338.5) |
| - | - | PubMed | | 6.36 (575.7) | 5.85 (348.1) |
| - | - | USPTO | + | 6.20 (491.0) | 5.63 (279.7) |
| | | PubMed | | | |
| GPT2-117M | 2.2B (1024L) | USPTO | | 5.86 (351.5) | 5.29 (197.8) |
| - | - | PubMed | | 5.82 (337.7) | 5.30 (200.0) |
| - | - | USPTO | + | 5.71 (301.4) | 5.15 (171.7) |
| | | PubMed | | | |
| GPT2-204M | 1.31B (1024 L) | USPTO | | 6.16 (473.4) | 5.40 (221.4) |
| - | - | PubMed | | 5.75 (314.2) | 5.25 (190.6) |
| - | - | USPTO | + | 5.60 (270.4) | 5.05 (156.0) |
| | | PubMed | | | |

- **Diversity coefficient increases with task batch size, but with diminishing returns.** Figure 8 (left) shows positive correlation between the diversity coefficient and batch size. This may be because larger batch sizes enable more unique tokens per batch.

- **Coefficients from using random probe networks likely diverge from using pre-trained networks.** Since the Task2Vec method (Achille et al., 2019a) uses a pretrained fine-tuned network, we consider the diversity computed using this configuration as a source of truth. Figure 8 (left) shows that using a random probe network underestimates diversity compared to pretrained networks, which is in accordance with results from Miranda et al. (2022b) on vision datasets and indicative of random networks' estimates diverging from pre-trained networks'.

- **Using a non-fine-tuned pre-trained network overestimates diversity.**

- Trends in diversity coefficient overestimation vs. underestimation for different probe network configurations are consistent across C4 and WikiText-103.

Based on these findings, we recommend using a batch size of 512 sequences for faster computations and fewer out of memory issues and pre-trained fine-tuned network. Since the other setting's diversity difference is large from the ground truth Achille et al. (2019a), we can't recommend it. If the

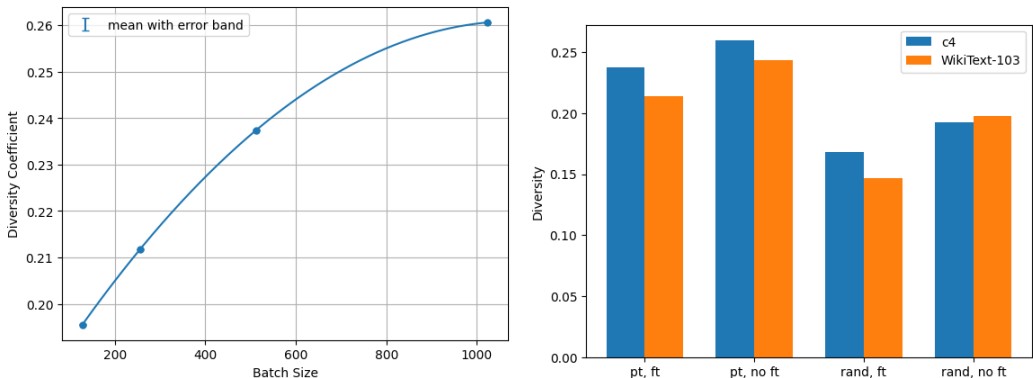

Figure 8: **Diversity coefficients of C4 computed using different task batch sizes show positive and diminishing returns with increasing batch size (left). Diversity coefficients of C4 and WikiText-103 computed using different GPT-2 probe network configurations show that random networks' estimates likely diverge from those of pretrained networks (in particular, they appear to underestimate diversity), and non-finetuned networks overestimate diversity vs. finetuned networks (right).** 95% confidence intervals for diversity coefficients are plotted, but are so small that they do not show. "pt" refers to pretrained network and "rand" refers to randomly initialized network. "ft" refers to a network that was finetuned per task and "no ft" refers to no finetuning performed.

intuitive properties reproduce for the other two options, we'd recommend it, but this is left for future work.

# I    EXPERIMENTAL DETAILS

## I.1    DATASET PREPROCESSING

In accordance with Achille et al. (2019a), we used the training split of datasets to finetune the probe network when computing Task2Vec embeddings per dataset. Sequences were tokenized using a pre-trained HuggingFace GPT-2 tokenizer based on byte-level Byte-Pair-Encoding, and padded or truncated to a max length of $128$. Because the WikiText-103 dataset contained empty text examples, we removed these examples before sampling batches to compute embeddings.

## I.2    MODEL ARCHITECTURE AND FINETUNING

We used a pre-trained GPT-2 model with a language modeling (LM) head on top. The pre-trained GPT-2 model itself has 12 layers, 12 heads, 768-d hidden size, and 117M total parameters. The LM head is a linear layer with weights corresponding to the input embedding layers. The model was pre-trained on the English language and the pre-trained GPT-2 tokenizer has a vocab size of $\approx 50k$ tokens. For all finetuning experiments, we fine-tuned only the LM head for 10 epochs. We used no learning rate scheduler and no gradient accumulation. We used the AdamW optimizer, since AdamW has been shown empirically to give better training loss and improved generalization.

We note that, in principle, the Task2vec diversity coefficient can be computed with any LLM. The metric itself is not specific to any particular LLM architecture or model version. We chose GPT-2 for our experiments due to computational efficiency and resource constraints. However, more powerful LLMs like LLaMA can also be used to compute the diversity coefficient. As long as the probe network used is consistent across experiments, the relative differences in the diversity coefficient value between datasets are directly comparable. The same goes for using pretrained vs. non-pretrained probe networks.

### I.3 NUMBER OF BATCHES AND BATCH SIZE SELECTION

Diversity coefficients in Table 1 were computed using randomly selected batches of size 512 sequences and a pre-trained, finetuned GPT-2 probe network. Diversity coefficients of C4, WikiText-103, The Pile, Pile-CC, HackerNews, NIH ExPorter, PubMed Abstracts, and USPTO were each computed using 200 sampled batches. Given resource constraints, we found 200 batches[3] to be a sufficiently large number of batches to estimate the diversity coefficient with tight 95% confidence intervals on the order of 1e-5. We chose 512 as the batch size, since it is a relatively large and feasible batch size to fine-tune the probe network on 200 batches using Azure NV12s_v3 instances equipped with Tesla M60 GPUs in a reasonable amount of time (30+ hours).

### I.4 DIVERSITY COEFFICIENT COMPUTATION OF CONCATENATED DATASETS

The diversity coefficient of a concatenated dataset of C4 and WikiText-103 was measured over a combined set of batches. Each batch consisted of sequences sampled from one of these datasets, e.g. a batch could have sequences randomly sampled from C4 or WikiText-103 but not both. The coefficient was computed over 400 batches of batch size 512 (200 batches from each dataset). Note that for the concatenated dataset, we utilized the same 200 batches per dataset that were used to compute the coefficients of C4 and of WikiText-103 individually.

The diversity coefficient of concatenated five sub-datasets of The Pile was computed over 1000 batches (200 batches from each dataset) of batch size 512. Similarly to the concatenated dataset of C4 and WikiText-103, we utilized the same 200 batches per dataset that were used to compute the coefficients of each individual sub-dataset.

### I.5 DIVERSITY COEFFICIENT OF THE PILE VS. CONCATENATION OF FIVE SUB-DATASETS

We make a clarification on the approach taken to evaluate the diversity coefficient for The Pile vs. for concatenation of its five sub-datasets.

The diversity coefficient of The Pile was computed over 200 batches sampled across all 22 sub-datasets of The Pile. This means that any given batch could contain sequences across all 22 sub-datasets, i.e. a batch could have sequences from Pile-CC, HackerNews, and NIH ExPorter.

The diversity coefficient of the concatenated dataset was computed over 1000 batches comprised of 200 batches separately sampled from each of the five sub-datasets. Each batch contained sequences from only one sub-dataset, i.e. a batch could only have sequences from Pile-CC or HackerNews or NIH ExPorter.

We hypothesize this distinction in the diversity coefficient computation explains why the concatenated dataset has higher diversity, even though it consists of only five of the 22 sub-datasets of The Pile. For the diversity coefficient of The Pile, because batches were sampled such that any batch contains sequences from across the 22 sub-datasets, the batch representations learned by the probe network may have been more similar, resulting in lower diversity relative to the concatenated dataset.

### I.6 DETAILS ON MIXES USED FOR INTERLEAVED DATASETS

"C4 and WikiText-103 (Mix1)" denotes an interleaved dataset, i.e. a dataset where examples from each sub-dataset are randomly mixed together according to their proportion in the mix, so that if you iterate through each example in a 1:1 mix then you might see e.g. Dataset1 example, Dataset2 example, Dataset1 example, Dataset1 example, Dataset2 example. Mix1 indicates that this interleaved dataset is 75% C4 examples and 25% WikiText-103 examples.

"Combination of five datasets (Mix2)" also denotes an interleaved dataset. Mix2 indicates that each sub-dataset constitutes X% of the interleaved dataset, where X% is chosen to as closely match LLaMAv1's data training mix as possible. For example, if LLaMAv1 is composed of twice as much web crawl data as Wikipedia data, then our Mix2 will have 2 Pile-CC examples per 1 WikiText-103

---

[3]This results in $(200^2 - 200)/2 = 19,900$ pairwise distances used to compute the diversity coefficient.

example.

## J PAIRWISE DISTANCE DISTRIBUTIONS OF C4, WIKITEXT-103, AND THE PILE

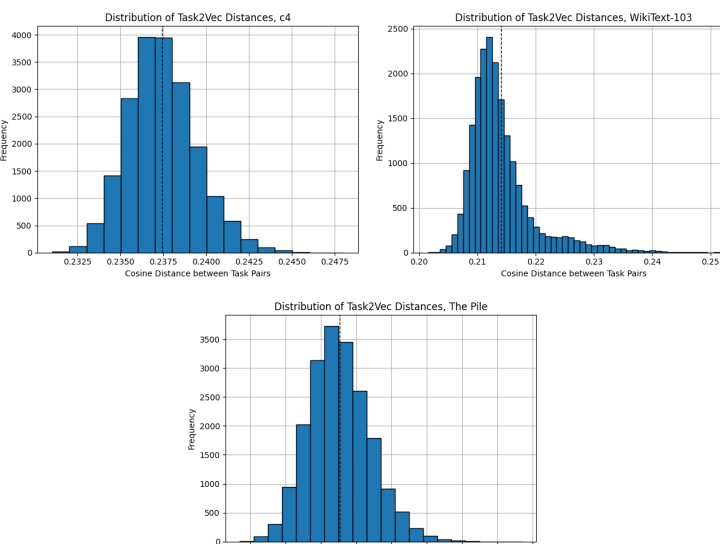

Figure 9: **Distributions of pairwise batch distances from C4 (top left), WikiText-103 (top right), and The Pile (bottom) are approximately Gaussian, which justifies the use of a sample of batches to measure the diversity coefficient.** Dotted lines indicate the average distance, i.e. the diversity coefficient, for each dataset.

**Experiments:** To provide confidence in the magnitude of the coefficient values of C4, WikiText-103, and The Pile, we plot the distribution of distances per dataset in Figure 9. We aim to show that a subsample of batches can provide a good estimation of population statistics, such as the diversity coefficient, which measures the expected Task2Vec (cosine) distance between batches.

**Results:** For each dataset, the pairwise distances take on unimodal and approximately Gaussian distributions with few outliers. These results suggest the Task2Vec distances are approximately normally distributed. This suggests we can make strong inferences about the population. Specifically, we are able to compute a good estimate of the diversity coefficient using 200 batches using the mean. This is in fact the same argument from Miranda et al. (2022a)—but we verified it applied in our setting. Figure 9 also shows few outlier batches—the presence of which could influence the computed diversity coefficient. This provides further confidence in the coefficient values computed and justifies our use of a sample of batches to estimate diversity.

**OpenWebtext:** Data from Reddit post URLs was extracted, deduplicated, and filtered for English content using FastText. Web pages were pulled using the newspaper python package, and near-duplicates were identified and removed using local-sensitivity hashing (LSH). Only documents with a unique content similarity of less than 0.5 and more than 128 tokens were retained. The finalized dataset comprises 38GB from 8,013,769 documents. Annotations: None present in the dataset. Used to train GPT2.

## K  GENERATIVE IN-CONTEXT LEARNING (GINC) DATASET

### K.1  BACKGROUND

The GINC dataset is generated using the latent concept framework proposed in Xie et al. (2021), where language models condition on a prompt to infer latent document concepts learned during pre-training. The pretraining distribution is defined using a uniform mixture of Hidden Markov Models (HMMs) parameterized over a family Θ of latent concepts.

### K.2  DEFINITIONS OF GINC DATASET PARAMETERS

**Number of latent concepts:** A latent concept $\theta$ parameterizes the transitions of a HMM in the mixture. A latent concept (e.g. a wiki bio) contains document statistics, such as semantics, syntax, and the formatting of and distribution of tokens.

**Vocabulary size:** Each HMM in a given mixture outputs a fixed number of tokens, defined as the vocabulary size. The vocabulary is generated by enumerating combinations of letters from a to z, aa to az, etc. The delimiter token is designated by a backslash. Sequences are tokenized by whitespace.

### K.3  SUPPLEMENTAL FIGURES FOR DIVERSITY COEFFICIENT VS. GINC PARAMETERS

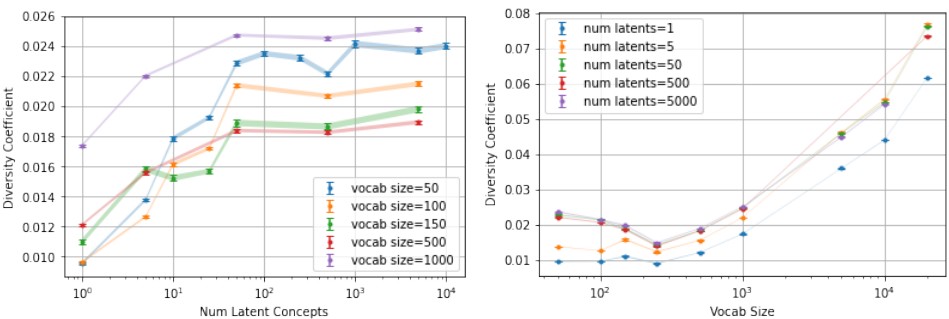

Figure 10: **Trends noted in Section 3.5 are consistent for diversity coefficient vs. number of latent concepts (left) and coefficient vs. vocab size (right) when the other parameter changes.** The diversity coefficient with 95% confidence intervals saturates with increasing number of latent concepts (left) even as vocab size is varied between 50-1000. Larger vocab sizes generally produce higher diversity coefficients (right) even as the number of latent concepts is varied between 1-5000.

Figure 10 confirms that the trends between the diversity coefficient and number of latent concepts (left) hold even as vocab size is varied. Similarly, trends between the diversity coefficient and the vocabulary size (right) hold as the number of latent concepts is varied. These trends were noted in Section 3.5.

## L  DISCUSSION (CONT.)

Our paper introduces a metric that leverages tunable parameters, such as the number of batches, batch size, probe network configuration (pre-trained vs. random, fine-tuned vs. not) and depth. While these elements influence the diversity coefficient's absolute value and necessitate the recalibration of lower and upper bounds (see Appendices I.3 and H), a consistent choice of hyperparameters can mitigate these effects.

Intriguingly, our proposed diversity may not always correlate with model performance, as high diversity could simply be due to uniform noise. Nevertheless, we contend that a higher diversity, in the context of a sufficiently large model, likely indicates superior performance and data quality. Furthermore, our diversity metric is intentionally designed to be widely applicable, albeit concealing causal factors, rendering it an effective tool for ablation studies.

Despite our diversity metric's broader applicability, it may obscure certain causal factors. This limitation is intentional to enhance its practical usage—since causality is often difficult to infer and is out of scope. This can be overcome with data property ablation studies, as we showed in our GINC dataset experiments.

Currently, our proposed bounds are specific to sequence data with a symbolic vocabulary, limiting their applicability across different modalities. To overcome this limitation, we suggest using a multimodal embedding method for embedding diversity coefficients and lower/upper bounds across tasks.

To really clarify why FIM is better than activations, we provide this intuitive explanation. FIM gives a weight/feature of which parameter of the generative distribution matters, e.g. the first coordinate of Task2Vec corresponds to how artsy the text sequence is. This is a feature of a task or dataset itself. Therefore, FIM exactly approximates the (task) data generative distribution we are trying to embed. Therefore, we conjecture it results in superior representations for datasets compared to activations since it directly approximates the data (or task) generative distribution. Our study, and references, provide positive evidence in favor of this argument.

The strength of embeddings is their ability to approximate **semantics** in a way that symbols may struggle with, such as distinguishing the equivalence of two sentences with different symbols but identical meanings. In NLP there is no easy way to determine this equivalence. In formal mathematics, symbolic semantics and thus equivalence can sometimes be done exactly. Though it does not come without its costs, e.g. requires expert knowledge, computationally demanding or (approximately) exhaustive representations like e-graphs. Therefore, embedding methods for data diversity, quality, etc. have the unique advantage of being more generally applicable.

Our diversity calculations predominantly utilize a small model (GPT-2). Despite the ongoing discussion concerning the emergence of large language models (LLMs), our conjecture extends the results to models of all sizes. We base this inference on the fact that the manifestation of emergence is intimately tied to the specific metric employed, and the sudden unpredictable jumps disappear when smooth metrics are applied Schaeffer et al. (2023b). The cosine distance is smooth and does not have this issue.

**Why and when does diversity matter?** We propose two central conjectures for the importance of diversity and provide the underlying rationale:

1. **Conjecture 1: Diversity is essential because it promotes learning-to-learn (a surrogate for General Intelligence).** The main argument is that a significant level of diversity corresponds to a multitude of tasks in the dataset. Therefore, to achieve high (test) performance, the model must perform well on all tasks. One potential strategy is by learning-to-learn, thereby allowing transferability when tasked with a new problem. Another alternative could be to memorize all tasks.

2. **Conjecture 2: Diversity is crucial because it enhances the probability that the pre-training set covers the test set.** Diversity is a formal score of coverage—it aims to reflect the effective number of tasks in a dataset. Thus, increased diversity equates to more tasks in a dataset. This (could) boosts the chance of the training set covering the test set, hence improving performance, given a sufficiently large model like an LLM. The direct exploration of this conjecture is slated for future investigation, but we provide a suggestive (correlative) analysis of one reason why LLMs might perform so well.

Another benefit is that our method does not rely on activations from an arbitrarily selected layer in a network. Lastly, note that activations may be unreliable for embedding dataset/tasks because large distances between datasets/tasks may be due to well-separated decision boundaries instead of intrinsic semantic properties of the dataset/task. In contrast, the diversity coefficient is well-justified, extensively tested in our work and previous work, e.g. the diversity coefficient correlates with ground truth diversities, cluster according to semantics, taxonomy etc. (see section F and Achille et al. (2019a); Miranda et al. (2022a)). In short, FIM-based representations are motivated by information theory (e.g. FIMs are metrics in distributions) and have been extensively tested by independent sources (Miranda et al., 2022a; Achille et al., 2019a; Vu et al., 2020).

Data has taken a central role in the success of modern machine learning methods, like GPT4 OpenAI (2023), CLIP Radford et al. (2021), and PaLM 2 Google (2023), with special relevance for architec-

tures with few inductive biases, like the popular Transformer (Vaswani et al., 2017). Therefore, it is paramount to understand the pretraining data we use, beyond scale alone. We conclude the diversity coefficient is a trustworthy metric, and conjecture the diversity coefficient can be used to build quality, diverse datasets for LLMs. We hope our contributions inspire more effective data collection and curation in machine learning that goes beyond scale alone to improve performance.

**Limitations:**

- The diversity coefficient presents an aggregate measure that masks the underlying causal factors. Despite this, we illustrate how it might be employed to uncover these factors. We show this through the use of vocabulary size and latent space, acknowledging that these experiments could be resource-intensive. Causality is a challenging topic, and we do not claim to solve it through our experiments. Our experiments in this regime are mostly to show that the diversity coefficient (might) correlates/captures different sources of diversity beyond number of concepts or tasks.

- The computation of Task2Vec embeddings requires more resources than computing simply the activations. However, given the proven correlation with ground truth task generative parameters from previous studies, we posit that it supersedes activations. Furthermore, we hypothesize that using activations could result in high distances due to optimization for decision boundaries, making it less reliable for measuring distances i.e., high distances in activation space might be artificially high. We observed this but plan to give more detailed study in the future.

- The choice of an expectation as the aggregation function could be seen as arbitrary. Alternatives such as the Vendi score are promising, but still under-explored and computationally demanding compared to expectations/sums. Future work could focus on the comparative analysis of the total distance sum and the Vendi score. We hypothesize, in line with the Central Limit Theorem (CLT), that the results might not differ significantly e.g., CLT still converges to (unit) Normal given the proper normalization procedure.

- We reject the notion that the use of models is a limitation. As discussed earlier, models can provide superior data embeddings, and all forms of data representations are always required. For example, the identity function or symbols are data representations.

**Implications:**

- Given the impressive performance of LLMs, our study suggests a correlation with our diversity measure, potentially providing an explanation for this high level of performance.

- High diversity implies a larger task coverage. Therefore, we conjecture that a highly diverse pre-training set could increase the probability of including relevant pre-training data for the target task/testing. This suggests that collecting more diverse data could be a method to enhance performance. If the model is sufficiently large, we conjecture this method always (stochastically) monotonically increases the performance (as implied by (Zhang et al., 2021)).

- The transition from a qualitative to a quantitative measure of diversity can be seen as a significant advancement in the field because of conceptual transitions about how we think and talk about data quality/diversity.

- The use of Task2Vec to embed data implies a method applicable to any modality, potentially benefiting all areas of machine learning research.

Importantly, the relationship between pre-training dataset diversity and test performance remains a key question. To investigate, we conduct preliminary experiments, pre-training three GPT-2 models on datasets of varying diversities, then evaluate their performance on diverse validation datasets. We observe in Table 2 that increased diversity leads to decreased cross-entropy loss, indicating a positive relationship between diversity and model performance on diverse tasks. Although the cross-entropy values are arguably large, Figure 7 shows that even the smallest 51M parameter models were still pre-trained enough to grasp important aspects of sentence syntax and structure of their respective dataset, indicating model performance on evaluation represents meaningful improvements in ability. Hence, we conjecture pre-training on higher diversity data improves test performance on diverse tasks, though more extensive experiments are needed to know so conclusively.

## L.1 FUTURE WORK

Our future research will explore the potential of the Task2Vec distance function for pre-training dataset curation. Given that the objective of pre-training is to maximize downstream task performance, we define high-quality training data as data that facilitates the best achievable performance on such tasks. We anticipate that higher diversity in the dataset will increase the likelihood of achieving this objective. The rationale is that a higher data diversity implies a broader coverage of tasks or batches, thereby increasing the probability of training the model on tasks or data representations that are relevant to evaluation tasks. Our focus will be to leverage Task2Vec to assess the similarity between individual data points, batches, or datasets to a target task. This assessment will enable us to curate the training data by selectively removing tasks that resemble random, noisy, or irrelevant sequences, which may adversely affect downstream performance.

