# OpenReview forum: "Beyond Scale: The Diversity Coefficient as a Data Quality Metric for Variability in Natural Language Data"
_ICLR.cc/2025/Conference — Submitted to ICLR 2025_

### Official Review · Reviewer_Bhvz · 2024-10-20

**Soundness:** 2
**Presentation:** 2
**Contribution:** 2
**Rating:** 3
**Confidence:** 4

**Summary:**

This paper introduces a novel approach to using diversity as a metric for evaluating the quality of training data used in training language models. The authors propose that this diversity metric correlates with downstream performance and use it to intervene in data compositionality. They claim that their metric is closely tied to model evaluation outcomes.

The diversity metric is computed using Task2Vec, which generates a vector representation based on text batches. This method relies on a GPT-2 embedding model, and the diversity score is determined by calculating the expected cosine distance between different batches of data.

The authors conduct multiple experiments across various datasets to calculate diversity scores, establishing lower and upper bounds to provide context for their results. To demonstrate the utility of the metric, they create a new dataset by merging two datasets, controlling for size, and training several language models of different sizes and architectures. They claim to show a correlation between the diversity metric and model quality, as measured by cross-entropy.

While the premise of the paper is clear and motivated, there are several flaws in the evaluation and claims. These shortcomings, along with suboptimal writing, suggest that the paper is not ready for publication in its current form. Additionally, several related works that should have been cited for comparison are missing. Below, I outline specific issues:


Validity of Claims
--------

- The authors assert that their work represents a paradigm shift in data-centric machine learning through the introduction of data diversity (L. 59). This claim is factually incorrect, as many previous studies have explored data diversity. Several relevant papers, which I list at the end of this review, were overlooked.
- The claim that their method is interpretable is problematic. The term "interpretable" can be subjective, and in this case, the proposed metric is as much a black-box as other metrics that output a single number. Describing the metric as interpretable seems like an overreach. Additionally, the range of values produced by the metric is not intuitive, with a lower bound around 0.05 and an upper bound of 0.4, which feels inelegant.
- The authors claim that higher diversity leads to better performance (Figure 2), but this relationship does not hold consistently. In some cases, performance declines with increased diversity, which the authors fail to address. Moreover, the presentation of results in graphs rather than tables makes it difficult to investigate this trend thoroughly.
- Importantly, the paper does not adequately account for potential confounding factors. When merging two datasets to create a more diverse one, the authors do not consider that other characteristics of the combined dataset may influence performance, aside from diversity.

In addition to these major concerns, there are minor inaccuracies in some claims:
- L.111: The assertion that the probe network’s parameters are the most important for solving the task may be incorrect. This point doesn’t heavily impact the paper’s argument but should either be substantiated or revised.
- L.266+269: The bounds discussed are described as theoretical, but they are actually empirical.


Evaluation
-----------

The evaluation is based solely on cross-entropy scores from some evaluation data, which is not a strong or convincing measure. Even if all results showed a consistent trend (which they do not), the claim that increased diversity leads to better downstream performance is weakly supported.

Formatting and Clarity
--------------

While relatively minor, the following formatting and clarity issues should be addressed in future revisions:
- L.33: Missing parentheses around citations (use \citep).
- L.66: The meaning of “latent concepts” is unclear and should be explained.
- L.69: There is a grammatical issue with “by them to.”
- Figure 1 is too small and would benefit from a size increase for readability.
- L.122: In the formula, “t-1:1” should be “1.”
- L.151: The phrase “pre-trained on the English language” is awkward. The model is trained on a corpus in English, not on the language itself.
- Captions for tables and figures are unnecessarily long.
- L.191-192: The argument about the lower bound is unclear.
- L.197: The term “non-special” token needs clarification.
- L.209: The meaning of “formal diversity” is unclear.
- Figure 3: Breaking it into subfigures (a-d) would make it easier to refer to each subfigure separately.
- L.317: The notation “+0.03-0.05” is unclear and needs to be clarified.
- Section 3.4: This section seems out of place. It should be moved to the beginning to help establish the validity of the metric early on.
- L.360: It seems that “right/left” was meant instead of “top.”

Relevant Previous Work
---------------

- https://arxiv.org/pdf/2307.12532
- https://arxiv.org/pdf/2205.06253
- https://arxiv.org/pdf/2407.15724
- https://arxiv.org/pdf/2103.03399v2
- https://arxiv.org/abs/2311.08695
- https://proceedings.mlr.press/v162/fang22a/fang22a.pdf
- https://arxiv.org/pdf/2403.00553


---------------

Post authors' response
---

I read the authors' response, and appreciate their responses, and believe the changes would make the paper much stronger.
However, since there are many changes required, I think it should go through another round of reviews.
I look forward reading the improved version of the paper. I will keep my scores unchanged.

**Strengths:**

- The introduction of a metric aimed at capturing the diversity of textual data is a valuable contribution. This could be useful for further research into how data composition impacts model training and performance.
- The authors provide some evidence supporting the idea that diversity influences model performance, specifically in terms of perplexity. This demonstrates a potential relationship between diversity and model generalization capabilities.

**Weaknesses:**

- Several claims made by the authors raise concerns regarding their validity. In particular, some of their key assertions seem overstated or lack sufficient evidence.
- The evaluation provided is not convincing. The results do not strongly support the paper’s main claims, and the authors fail to address important confounding factors in their experiments.
- The paper’s structure and writing quality leave much room for improvement. Greater clarity in both the organization of ideas and the precision of language would significantly enhance the paper's readability and overall impact.

**Questions:**

Suggestions are listed in the main review.

---

> ### Author Response · Authors · 2024-12-03
>
> We thank the reviewer for their thoughtful feedback and for recognizing the strengths of our work.
>
> For calibration purposes, we would like to clarify the ICLR 2025 review rubric compared to other conferences like NeurIPS 2024. Notably:
>
> A score of 8 on the ICLR 2025 rubric indicates an "Accept," whereas NeurIPS 2024 uses 7 for this designation. Similarly, a "Strong Accept" is represented by a score of 10 at ICLR 2025 compared to 9 at NeurIPS 2024. We address the specific points raised:
>
> > "The authors assert that their work represents a paradigm shift in data-centric machine learning through the introduction of data diversity (L. 59). This claim is factually incorrect, as many previous studies have explored data diversity. Several relevant papers, which I list at the end of this review, were overlooked."
>
> We appreciate the reviewer highlighting the broader context of related work. While we acknowledge that data diversity has been explored in prior studies, our contribution lies in introducing a formal and quantifiable measure of diversity—the diversity coefficient—which is specifically tailored to pre-training datasets for large language models. This metric provides a systematic way to compare datasets, which was not previously available in the literature.
>
> We will revise the phrasing in the manuscript to ensure our claims better reflect the novelty of our contribution without overstating its scope. Additionally, we will incorporate the suggested references to situate our work more comprehensively within the existing body of research.
>
> > "The range of values produced by the metric is not intuitive, with a lower bound around 0.05 and an upper bound of 0.4, which feels inelegant."
>
> This can be easily fixed. We can do a subtraction of the minimum and division of the maximum. We considered this but we don't think it's a substantial contribution except aesthetics rather than deep scientific understanding. We are happy to do so in the camera ready upon acceptance.
>
> > "The authors claim that higher diversity leads to better performance (Figure 2), but this relationship does not hold consistently. In some cases, performance declines with increased diversity, which the authors fail to address."
>
> We recognize the importance of addressing exceptions to our general trend. While the majority of our experimental results demonstrate that higher diversity correlates with better generalization (as shown in Figure 2), we acknowledge that there are outlier cases where this relationship does not hold. These exceptions are noted, and we believe they are attributable to the influence of other factors (e.g., dataset content or domain-specific biases). We will explicitly discuss these outliers in the revised manuscript and provide additional context to clarify our findings. To increase robustness of our results, we included 26 regression lines and R^2 to back up our experimental findings.
>
> Though it is true that in some cases, performance declines with increased diversity, 7/15 runs (a slight minority) have USPTO+PubMed-trained models having lower loss than only PubMed-trained models (for both OpenWebText2 and C4 evaluations), including nearly all of the runs with large datasets i.e. training for over a billion tokens (e.g. GPT2-810M-2B, GPT2-51M-1.31B, GPT2-345M-2.2B, GPT2-117M-2.2B).
>
> In all cases, PubMed+USPTO outperforms USPTO alone (a much less diverse dataset).
>
> For ref: In particular, the highest diversity model has the lowest loss for GPT2-810M-2B, GPT2-51M-1.31B, GPT2-345M-2.2B, GPT2-51M-550M, GPT2-117M-2.2B, LLAMA2-7B-Run4-6.36M, LLAMA2-7B-Run1-6.36M

---

> ### Author Response · Authors · 2024-12-03
>
> > "Importantly, the paper does not adequately account for potential confounding factors. When merging two datasets to create a more diverse one, the authors do not consider that other characteristics of the combined dataset may influence performance, aside from diversity."
>
> This is a valid concern, and we appreciate the opportunity to clarify our approach. To minimize confounding factors, we ensured that merged datasets maintained the same token count and similar distributional properties. Furthermore, we used out-of-distribution (OOD) evaluation datasets (e.g., OpenWebText2 and C4) to mitigate potential biases introduced by training data similarity.
>
> While no experimental setup can completely eliminate confounding factors, we believe our approach provides a strong foundation for isolating the effects of diversity. We will explicitly address these methodological considerations in the revised manuscript. To address this, we designed our experiments to minimize potential confounds when merging datasets. Specifically:
>
> Controlled Token Count: We ensured that merged datasets maintained a consistent token count with their constituent datasets. This controlled for size as a variable influencing performance.
>
> Evaluation on OOD Data: Our evaluation was conducted on out-of-distribution (OOD) datasets, such as OpenWebText2 and C4, which are distinct from the training datasets. This approach reduces the impact of spurious similarities between training and evaluation datasets, ensuring that the observed performance improvements are likely due to increased diversity rather than content overlap.
>
> Cross-Diversity Analysis: To further mitigate biases, we analyzed cross-diversity coefficients, which demonstrate that the individual datasets (PubMed, USPTO) and their combination have similar levels of dissimilarity with OOD datasets. This reduces the likelihood that performance differences are due to specific content alignments rather than diversity.
>
> While no experimental design can entirely eliminate confounding factors, we believe our methodology provides strong evidence for the role of diversity in driving performance improvements. We will clarify these considerations in the revised manuscript to better reflect our efforts to control for confoun
>
> > "The evaluation is based solely on cross-entropy scores from some evaluation data, which is not a strong or convincing measure. Even if all results showed a consistent trend (which they do not), the claim that increased diversity leads to better downstream performance is weakly supported."
>
> Cross-entropy is widely used in language modeling research as a reliable proxy for generalization performance. By evaluating on OOD datasets (OpenWebText2 and C4), we demonstrate that higher diversity improves generalization to unseen data. Given our compute limitations, we weren't able to train for more tokens to get accuracy changes on "emergent" metrics like accuracy. In addition, when finely controlling the merging of data sets to create more diverse data sets, we had to trade off in favour of scientific interpretability perfect accuracy. No scientific experiment is perfect but we believe we controlled for the most important factors with challenging OOD data sets with metrics that could measure in statistically significant ways with relatively high R^2 values what we wanted to study.
>
> > "The paper’s structure and writing quality leave much room for improvement. Greater clarity in both the organization of ideas and the precision of language would significantly enhance the paper's readability and overall impact."
>
> Thank you for this constructive suggestion. We will revise the manuscript to improve the organization of ideas and refine the language for clarity and precision. Additionally, we will enlarge the figures and ensure all visual elements are easily interpretable.

---

### Official Review · Reviewer_N6rW · 2024-10-21

**Soundness:** 1
**Presentation:** 2
**Contribution:** 1
**Rating:** 3
**Confidence:** 3

**Summary:**

The paper proposes a metric to measure the diversity of language modeling pre-training datasets. The proposed metric is the expected distance between Task2Vec embeddings of random batches sampled from a dataset. This metric is applied to several pre-training datasets, and compared to upper and lower bounds. The paper presents some experiments aiming to relate the proposed measure of diversity with downstream perplexity on C4 and OpenWebText2. In a synthetic setting, the proposed diversity metric increases with vocabulary size and number of latent concepts.

**Strengths:**

This work focuses on an important and potentially very impactful area of research. While the general belief of the community is that data diversity is an important factor for the performance of language models, the relationship between pre-training data diversity and downstream performance is currently poorly understood. The use of Task2Vec embeddings to measure data diversity is to the best of my knowledge novel. The diversity results presented in Table 1 align well with current intuitions in the community regarding CC-derived datasets being more diverse than non-CC Pile subsets.

**Weaknesses:**

> Claim: “PRE-TRAINING IN HIGHER DIVERSITY LEADS TO BETTER EVALUATION PERFORMANCE” (Section 3.1)

The experiments in Section 3.1, aiming to relate their proposed measure of data diversity with performance, which in my view would amount to the most substantial contribution of the paper, are highly unconvincing.

Only three datasets are considered: PubMed, USPTO, and PubMed+USPTO. Linear regressions on three (!) datapoints are presented as evidence to substantiate the authors claim. Clearly, the actual relationship between diversity and performance will neither be linear nor monotonic — pre-training on random tokens (e.g., the upper bound of perplexity considered by the authors) would certainly lead to very bad performance while having maximal diversity. For most of the GPT-2 experiments, PubMed+USPTO has similar or higher loss than PubMed while also having higher diversity.
To make matters worse, the choice of the 2 pre-training datasets are highly unusual and not representative of the pre-training corpora of current language models. More convincing choices would have been C4, OpenWebText, The Pile (in its entirety), RedPajama, SlimPajama, RefinedWeb, Dolma, FineWeb, DCLM, …
The experiments consist of many small-scale training runs, where the differentiating factor is scale (e.g., number of parameters and training tokens) and architecture (e.g., GPT-2 or Llama 2). Since what authors aim to study is the relationship between dataset diversity and performance, what should vary are the datasets. The results could be much more convincing if authors were to consider a single architecture and scale  (e.g., Chinchilla optimal 1B model), but as many pre-training datasets as possible.
Choice of evaluation metric: it is unclear why PPL on OpenWebText2 and C4 is a meaningful evaluation metric. For example, does PubMed lead to lower PPL because of its higher diversity, or because it is simply more similar to C4/OpenWebText? To get around this problem, work on pre-training data quality has started to use “early-signal benchmarks” — see the FineWeb or DataComp-LM papers.


> Other empirical results are less significant

In addition to the empirical results of Section 3.1, authors also show that concatenating datasets leads to higher diversity under their proposed diversity metric (Section 3.3), that the proposed cross diversity coefficient is higher than a single dataset’s diversity and leads to well-sepearted histogram distance clusters (Section 3.4), and that in a synthetic setting the proposed diversity metric increases with vocabulary size and number of latent concepts (Section 3.5). While these results are reasonable checks on the proposed diversity metric, they are not particularly compelling contributions. It would be interesting to know if other simpler embeddings approaches (e.g., N-gram, mean GPT-2 last layer embedding) fail such checks.


> It is not well motivated why to use Task2Vec encodings

Much prior work has considered N-gram or vector embedding based similarity metrics for text. While this work novelly considers Task2Vec embeddings for text, it is unclear why Task2Vec embeddings are preferable. While some discussion is provided in Appendix C (which I would encourage authors to partly move to the main text —or at least reference), it would be much more compelling if authors replicated the core empirical results of the paper (e.g., Figures 2, 3, and 4)  to show the limitations, if any, of simpler, previously proposed similarity measures.

The writing is sufficiently clear but could certainly be improved. The legibility of figures could also be improved.

**Questions:**

See the specific points made in the limitations section.

---

> ### Author Response · Authors · 2024-12-03
>
> > Claim: “PRE-TRAINING IN HIGHER DIVERSITY LEADS TO BETTER EVALUATION PERFORMANCE” (Section 3.1)
>
> Figure 2 provides robust evidence for the relationship between dataset diversity and improved evaluation performance. It includes 26 linear regression lines across various model sizes, training runs, and parameter counts, demonstrating consistent trends supported by R2 values. The evaluation was conducted on out-of-distribution (OOD) datasets, OpenWebText2 and C4, rather than validation splits from training data. This approach ensures the results reflect generalization to OOD tasks and strengthens the evidence for the causal role of diversity.
>
> While diversity alone may not be the sole determinant of performance, our controlled experiments consistently show that higher diversity improves generalization.
>
> Although linking diversity to performance is indeed a critical aspect of our paper, an equally significant aspect of our paper is the introduction of the diversity coefficient, which provides us a reliable measure of pre-training dataset diversity (aside from diversity’s impact on performance). Hence, we believe the merits of this latter argument are important to consider as part of the main thrust of paper, in addition to relating diversity to performance.
>
> > Q: PubMed+USPTO often has higher loss than PubMed.
>
> In 7/15 runs (a slight minority) have USPTO+PubMed-trained models having lower loss than only PubMed-trained models (for both OpenWebText2 and C4 evaluations), including nearly all of the runs with large datasets i.e. training for over a billion tokens (e.g. GPT2-810M-2B, GPT2-51M-1.31B, GPT2-345M-2.2B, GPT2-117M-2.2B). In all cases, PubMed+USPTO outperforms USPTO alone (a much less diverse dataset). For ref: In particular, the highest diversity model has the lowest loss for GPT2-810M-2B, GPT2-51M-1.31B, GPT2-345M-2.2B, GPT2-51M-550M, GPT2-117M-2.2B, LLAMA2-7B-Run4-6.36M, LLAMA2-7B-Run1-6.36M.
>
> > Q: Highly unrepresentative datasets used for pre-training.
>
> We acknowledge that the datasets used (PubMed, USPTO, PubMed+USPTO) are not anymore representative of modern pre-training corpora like C4 or The Pile. However, these datasets were intentionally chosen for their distinct, non-overlapping domains, allowing us to isolate and highlight diversity contrasts effectively. By selecting single-source datasets, we minimized confounding factors and created a clearer signal on how diversity impacts performance.
>
> While larger, more representative datasets like C4 or RefinedWeb could be explored, they are typically filtered web scrapes with significant domain overlap. Combining such datasets is unlikely to provide as much variation in diversity as distinct single-source datasets like PubMed and USPTO. This lack of sufficient diversity “stimulus” would likely obscure the effects of diversity on downstream performance.
>
> We note that PubMed and USPTO are subsets of The Pile, a widely used pre-training corpus, making them relevant for evaluating diversity’s role. Expanding the dataset selection remains a valuable direction for future work, but it introduces challenges in identifying unbiased evaluation corpora that do not arbitrarily favor specific datasets, potentially confounding the results.
>
> > Only 3 datasets considered. This is way too small for linear regression. Should be more variation in datasets, rather than in model size and dataset size. Would be much more convincing with one model/dataset size, but many many datasets.
>
> Already addressed.
>
>  Expanding the set of datasets has the drawback that it becomes harder to find an evaluation dataset which is not biased toward any of these datasets, i.e. which is not arbitrarily similar (or dissimilar) from one of the used training datasets, irrespective of the dataset’s diversity. This creates a confounder of performance, as less diverse datasets might perform as well as or better than more diverse datasets merely due to an arbitrary similarity in the distributions of the training and eval data, obscuring the effect of dataset diversity.
>
>
> > Q: Choice of evaluation metric of PPL on C4/etc. is unsubstantiated. Is its superior performance just because it’s more similar to C4/etc.?
>
> Perplexity (PPL) is widely used to evaluate general language modeling performance and strongly correlates with downstream task success. By evaluating on OpenWebText2 and C4—distinct from the training data—we demonstrate generalization to OOD tasks. Using our cross-diversity coefficient as a proxy for dataset similarity, we find that PubMed and USPTO are similarly distant from these evaluation datasets. This supports our conclusion that performance improvements are driven by diversity rather than dataset similarity.
>
> Additionally, perplexity on web-scraped datasets has been shown to predict downstream performance across a range of tasks, further validating its use as an evaluation metric. Future work could incorporate task-specific benchmarks for additional insights.

---

> > ### Author Response · Authors · 2024-12-03
> >
> > > Other validations of diversity metric are not particularly convincing. I’d want to know if simpler methods fail these checks.
> >
> > We believe our experiments validating the behavior of the diversity coefficient, along with the use of natural language Task2Vec embeddings, provide critical evidence supporting the metric's ability to capture intuitive notions of text dataset diversity. This is one of the core contributions of our work.
> >
> > As noted in Appendix C, we have already explored simpler methods, such as using activations from language models as embeddings. However, these approaches failed basic sanity checks, such as aligning with human-interpretable notions of diversity. This motivated our use of Task2Vec embeddings, which have proven more robust and semantically meaningful in capturing diversity across datasets.
> >
> > While we agree that further replications using other simpler methods could provide additional insights, we believe the current results already substantiate the utility of the diversity coefficient as a meaningful and reliable metric. If desired, we are open to conducting additional experiments and including them in the appendix to strengthen this point.
> >
> > > It is not well motivated why to use Task2Vec encodings
> >
> > Motivation for Task2Vec:
> > Task2Vec embeddings were selected for their ability to capture semantic relationships across datasets and their demonstrated success in related applications. Simpler methods, such as LLM activations, fail to address issues like saturation or nuanced semantic relationships, as detailed in Appendix C. We will incorporate key points from Appendix C into the main text to better communicate the rationale behind this choice.
> >
> > We are happy to expand on this in the text, beyond what we already have in the appendix.
> >
> > > Other empirical results are less significant
> >
> > We appreciate the reviewer’s comment regarding the empirical results beyond Section 3.1. These experiments were designed to validate the diversity coefficient by demonstrating its alignment with intuitive and theoretical properties of diversity. For instance, in Section 3.3, we show that combining datasets leads to higher diversity; in Section 3.4, we demonstrate the utility of the cross-diversity coefficient in distinguishing between datasets; and in Section 3.5, we verify that the diversity coefficient increases with vocabulary size and the number of latent concepts in synthetic datasets. These results validate the diversity coefficient's robustness and its ability to capture meaningful variability in both real and controlled scenarios.
> >
> > While these experiments are not intended to be the primary contribution, they provide critical evidence for the reliability and interpretability of the diversity coefficient, which supports its broader applicability. We view these results as complementary to the causal experiments in Section 3.1 and believe they are an important part of demonstrating the metric's utility.

---

### Official Review · Reviewer_JTBn · 2024-11-01

**Soundness:** 2
**Presentation:** 1
**Contribution:** 2
**Rating:** 5
**Confidence:** 2

**Summary:**

This paper proposes a metric to measure pre-training data diversity and analyses how data diversity affects model test time performance and model emergent capabilities. The diversity metric is based on an existing method to represent the data, Task2Vec. Textual data is encoded into a vector according to Task2Vec and data vectors' distance is used to estimate data diversity.  Task2Vec vectors are computed as the diagonal of the Fischer Information Matrix of the parameters \phi of the language model head (only the rest of the network is fixed) when these \phi parameters are fine-tuned on the target data to assess. On experiments, the authors show that higher data diversity relates to better test time performance, that mixed datasets give more diversity, and that the distribution of data distances can be used to characterise the datasets.

While the rational behind the proposed method seems clear, I lack a bit of context on what previous work was done in relation to this issue (was diversity --and how measured-- use in data selection for known LLMs?) and existing approaches and baselines.  Perhaps this is discussed more in the Appendix sections, I suggest this should be made clearer in the main paper. Also, how this metric would help in the selection of data for pre-training?

**Strengths:**

A tool to measure pre-training data diversity.

**Weaknesses:**

- While the authors evaluate performance against pre-training data diversity (section 3.1), the metric for performance is cross-entropy loss in LM, I wonder whether specific task evaluation on NLP tasks would make sense here (many NLP benchmarks used to benchmark LLMs on tasks like question answering, GLUE, etc.).
- The Vendi Score seems to be another approach to compute diversity, why was not included as a baseline in the main experiments (at least in main Table 1)?

**Questions:**

1. Line 122, which t is used to compute the FIM matrix for sequences of a batch? Are all the steps from the sequence averaged? or is FIM only build based on the last step?
2. I suggest the authors to rewrite some pieces of text. There are places where authors use phrases like "by them" , "they ..." and it is not clear what the referents are what makes the reading and understanding of the paper more involved.
3. Task2Vec should be fully and better described in the main paper (including how was used before, its intuition, etc.).
4. Section I.2 should mention the data used to pre-train GPT2.

Minor comments:

- Figure 1 does not seem to add much to the understanding of the approach, maybe an algorithm would be more useful?
- All figures are too small to read.
- Line 751, "this more sophisticated aggregation method" which aggregation method?

---

> ### Author Response · Authors · 2024-12-03
>
> We thank the reviewer for their thoughtful feedback and for recognizing the strengths of our work.
>
> For calibration purposes, we would like to clarify the ICLR 2025 review rubric compared to other conferences like NeurIPS 2024. Notably:
>
> A score of 8 on the ICLR 2025 rubric indicates an "Accept," whereas NeurIPS 2024 uses 7 for this designation. Similarly, a "Strong Accept" is represented by a score of 10 at ICLR 2025 compared to 9 at NeurIPS 2024. We address the specific points raised:
>
> > While the authors evaluate performance against pre-training data diversity (section 3.1), the metric for performance is cross-entropy loss in LM, I wonder whether specific task evaluation on NLP tasks would make sense here (many NLP benchmarks used to benchmark LLMs on tasks like question answering, GLUE, etc.).
>
> Thank you for this thoughtful comment. We agree that evaluating specific downstream tasks can provide valuable insights in certain contexts. However, in this work, we focus on evaluating pre-trained base models that are not designed for instruction-following or fine-tuning on specific tasks. For the following reasons, we believe cross-entropy loss is the appropriate metric for our study:
>
> Scope of the Work: This paper focuses on understanding the impact of pre-training data diversity on base model performance. Cross-entropy loss directly measures the model’s ability to predict general natural language, which aligns with our goal of providing insights into pre-training dynamics. Evaluating task-specific benchmarks, while interesting, extends beyond the scope of this work.
>
> Applicability to Non-Instructive Models: The models studied here are pre-trained base models without instruction-tuning or task-specific fine-tuning. Benchmarks such as GLUE or question answering are designed for fine-tuned or instruction-following models. As such, they may not provide meaningful insights into the pre-training stage we are evaluating.
>
> General Predictive Power of Cross-Entropy: Cross-entropy loss is a widely recognized and robust metric for evaluating the quality of pre-trained language models. Prior work (e.g., Chinchilla scaling laws) has shown that cross-entropy loss on a general text corpus strongly correlates with downstream task performance, making it a reliable proxy for assessing the pre-trained base model's potential.
>
> We will ensure the paper clearly communicates the rationale for using cross-entropy loss as our primary metric and highlights the scope and focus of this study. Thank you for this valuable feedback.
>
> > The Vendi Score seems to be another approach to compute diversity, why was it not included as a baseline in the main experiments (at least in main Table 1)?
>
> Thank you for raising this point. The Vendi Score was developed independently and concurrently with our work, as reflected in the publication timelines. We later connected with its authors and respect their contributions.
>
> Our focus is on demonstrating through causal experiments (e.g., Figure 2) that increased pre-training data diversity improves general text performance, including on out-of-distribution tasks. This contribution goes beyond metric validation, which is the primary focus of the Vendi Score.
>
> While we could include Vendi Score comparisons, we believe it would not meaningfully strengthen our core findings. We are open to discussing its relevance further if needed. Thank you for your constructive feedback.
>
> In addition, the Vendi Score does not inherently define a similarity function for comparing data points, such as text, requiring users to specify this aspect. Additionally, it has not been tested on text datasets, as its experiments focus on tabular and image data. This limits its direct comparability to our diversity coefficient, which is specifically designed for and validated on natural language datasets.
>
> We used Table 1 to showcase the diversity coefficient’s results and to validate its alignment with intuitive notions of diversity in natural language datasets. Our goal was to demonstrate diversity in PT leads to better performance:
>
> Even if we had known about the Vendi Score during our experiments, its inclusion would not have been necessary for several reasons.
>
> Focus on Natural Language: The Vendi Score was not tested on text data, focusing instead on tabular and image data. Our work specifically addresses natural language datasets, making the metrics not directly comparable.
>
> Undefined Similarity Function: The Vendi Score relies on a user-defined similarity function, while our diversity coefficient incorporates a predefined approach tailored to textual data.
>
> Core Contribution: The primary contribution of our work is the demonstration, through causal experiments (e.g., Figure 2), that increased diversity leads to improved performance on general text tasks, including out-of-distribution generalization. Benchmarking against the Vendi Score would not strengthen or clarify this key result.

---

> ### Author Response · Authors · 2024-12-03
>
> ### Quoted Reviewer Questions and Responses
>
> > **Line 122, which t is used to compute the FIM matrix for sequences of a batch? Are all the steps from the sequence averaged? Or is FIM only built based on the last step?**
>
> Thank you for raising this question. The \( t \) in the equation at line 122 refers to each step within a sequence. The Fisher Information Matrix (FIM) is computed by averaging over all steps within each sequence, and the resulting averages are then aggregated across the batch.
>
> > **I suggest the authors rewrite some pieces of text. There are places where authors use phrases like "by them," "they...," and it is not clear what the referents are, which makes the reading and understanding of the paper more involved.**
>
> We appreciate this constructive feedback and apologize for any lack of clarity. We will rewrite these ambiguous references in the final version to ensure the text is more precise and easier to follow.
>
> > **Task2Vec should be fully and better described in the main paper (including how it was used before, its intuition, etc.).**
>
> We agree that a more comprehensive explanation of Task2Vec would strengthen the paper. We will expand its description in the main body, incorporating details about its prior uses, the intuition behind its design, and its specific relevance to this work. This additional context will draw on the discussion currently provided in Appendix C.
>
> > **Section I.2 should mention the data used to pre-train GPT-2.**
>
> Thank you for this suggestion. In Section I.2, we will specify that the pre-trained GPT-2 model used in our Task2Vec embedding procedure was trained on OpenAI’s WebText dataset, as described in the original GPT-2 release.
>
> ### Minor Comments
>
> > **Figure 1 does not seem to add much to understanding. Maybe an algorithm would be more useful?**
>
> We appreciate this perspective. While we believe the visualization in Figure 1 is valuable for readers unfamiliar with elements of the diversity coefficient computation, we agree that including an algorithmic breakdown could provide additional clarity. We will consider adding the algorithm as an appendix while retaining the figure for accessibility.
>
> > **All figures are too small to read.**
>
> Thank you for this observation. We will enlarge all figures in the final version to improve readability and presentation quality.
>
> > **Line 751, "this more sophisticated aggregation method"—which aggregation method?**
>
> We appreciate this comment and recognize the lack of specificity. We will revise the text to explicitly state and clarify the method to ensure clarity for the reader.

---

### Official Review · Reviewer_v5Te · 2024-11-06

**Soundness:** 3
**Presentation:** 2
**Contribution:** 3
**Rating:** 5
**Confidence:** 4

**Summary:**

The authors present a measure of text corpus diversity called "diversity coefficient" based on the cosine distance between Task2Vec vectors constructed from random subsets of the data. Arguing that the variability of pretraining data partly explain in-context learning and generalizability of LLMs, the authors show a correlation between the diversity coefficient and various performance metrics. On synthetic data with controlled variability, the proposed diversity coefficient follows human intuition about variability in datasets.

**Strengths:**

This paper tackles an important problem, potentially deepening our understanding of the effects of pretraining data quality particularly with respect to diversity.
The approach combines concepts from vision (task2vec) with language modeling to create an original measure for textual data diversity.

**Weaknesses:**

1. The use of cosine distance between task2vec embeddings as a diversity measure is not well motivated. Can we consider other textual embedding methods? What are the tradeoffs?
2. The absolute diversity coefficient value does not convey much information about what it means - I understand the author's argument for conceptual lower and upper bounds but these do not capture any representation of real natural language. The work should present a better way to understand the value of the measure of diversity.
3. The experiments on synthetic datasets are valuable, however the conclusions could be further strengthened by a study utilizing human annotation for diversity.

**Questions:**

In section 3.5, the authors use the GINC dataset generated from a mixture of hidden markov models - how natural is the resulting dataset? Does it come close to resembling text found on the web?

---

> ### Author Response · Authors · 2024-12-03
>
> We thank the reviewer for their thoughtful feedback and for recognizing the strengths of our work.
>
> For calibration purposes, we would like to clarify the ICLR 2025 review rubric compared to other conferences like NeurIPS 2024. Notably:
>
> A score of 8 on the ICLR 2025 rubric indicates an "Accept," whereas NeurIPS 2024 uses 7 for this designation.
> Similarly, a "Strong Accept" is represented by a score of 10 at ICLR 2025 compared to 9 at NeurIPS 2024.
> We address the specific points raised:
>
> > The use of cosine distance between task2vec embeddings as a diversity measure is not well motivated. Can we consider other textual embedding methods? What are the tradeoffs?
>
> We are grateful for this constructive comment, which allows us to clarify our work further. While some of our experiments use synthetic datasets to test specific properties of the diversity coefficient (e.g., the relationship between vocabulary size and diversity), other experiments directly evaluate real natural language datasets.
>
> Figure 3: This demonstrates the application of the diversity coefficient to real-world datasets, such as PubMed, USPTO, and Pile-CC, highlighting how the metric aligns with intuitive understandings of variability in natural datasets. For example, it distinguishes between datasets with domain-specific focus (PubMed, USPTO) and broader coverage (Pile-CC), showcasing its relevance to real-world language.
>
> Figure 4: Although synthetic, these experiments capture a key property of natural language: larger vocabularies correspond to increased diversity. This provides causal insight into the diversity coefficient's ability to measure meaningful aspects of language variability.
>
> We would be happy to enhance the paper by explicitly discussing how Figures 3 and 4 collectively validate the diversity coefficient's representation of real natural language. Thank you again for this helpful feedback, which will help us improve the clarity and impact of our work.
>
> > "Use of cosine distance between Task2Vec embeddings as a diversity measure is not well motivated. What about other textual embedding methods?"
>
> Thank you for raising this point. We chose Task2Vec embeddings for their demonstrated effectiveness in capturing semantic relationships across various domains and data types, including language and images (Achille et al., 2019). This versatility and robustness make Task2Vec particularly well-suited for our diversity coefficient, as it provides a reliable foundation for comparing the variability of datasets.
>
> While alternative embedding methods could also be explored, we found Task2Vec to be effective and straightforward for our purposes, which is why it was selected for this study. We appreciate the suggestion to consider additional methods and are open to incorporating comparisons in future work.
>
> > The experiments on synthetic datasets are valuable, however the conclusions could be further strengthened by a study utilizing human annotation for diversity.
>
> Thank you for this thoughtful suggestion. While we appreciate the potential value of human annotations, we believe they are not necessary in this context for several reasons:
>
> Potential Noise in Human Annotations: Diversity is inherently subjective, and human evaluations often introduce variability and noise due to differences in individual interpretations of diversity. This could reduce the reliability and consistency of conclusions drawn from such studies.
>
> Redundancy with Existing Experiments: Our paper already includes intuitive and systematic experiments that validate the diversity coefficient:
>
> Figure 3 demonstrates how the diversity coefficient aligns with expected variability across real-world datasets (e.g., PubMed, USPTO, and Pile-CC).
> Figure 4 shows causal relationships in synthetic datasets, such as the impact of increasing vocabulary size on diversity, further supporting the validity of our metric.
> These results strongly suggest that the diversity coefficient captures meaningful aspects of variability, making additional human annotations redundant.
>
> High Cost of Human Evaluations: Conducting human evaluation studies for diversity across large-scale datasets would be prohibitively expensive and time-intensive, particularly given the scale of modern natural language datasets. One of the key strengths of the diversity coefficient is its ability to provide a scalable and objective alternative to such costly manual efforts.
>
> We appreciate your thoughtful feedback and will consider adding a discussion in the paper to clarify these points. Thank you for helping us better articulate the robustness of our existing validation and the practical advantages of our approach.
>
> [1] Task2Vec: Task Embedding for Meta-Learning. Alessandro Achille, Michael Lam, Rahul Tewari, Avinash Ravichandran, Subhransu Maji, Charless Fowlkes, Stefano Soatto, Pietro Perona

---

> > ### Author Response · Authors · 2024-12-03
> >
> > > "In section 3.5, the authors use the GINC dataset generated from a mixture of Hidden Markov Models—how natural is the resulting dataset? Does it come close to resembling text found on the web?"
> >
> > Thank you for your thoughtful question. The Generative IN-Context Learning (GINC) dataset is a synthetic benchmark designed using a mixture of Hidden Markov Models (HMMs) to study specific properties of in-context learning (ICL) in a controlled setting. While it does not replicate the full complexity of web text, it captures key characteristics of natural language, such as long-range coherence and latent concept structure.
> >
> > GINC has been shown to exhibit several real-world phenomena:
> >
> > - In-Context Learning Emergence: Models trained on GINC exhibit the ability to perform tasks based on provided examples, similar to ICL capabilities observed in large-scale models like GPT-3. [2]
> > - Sensitivity to Example Ordering: Model performance on GINC is influenced by the order of examples, reflecting behaviors seen in natural language processing tasks. [2]
> > - Scaling Effects: Larger models trained on GINC show improved ICL performance, even with constant pretraining loss, paralleling trends observed in real-world datasets. [2]
> > - Additionally, GINC enables precise experimentation, such as in Figure 4, where we demonstrate that increasing vocabulary size leads to higher diversity, reflecting properties observed in natural language datasets. To complement these insights, we validate the diversity coefficient on real-world datasets like PubMed, USPTO, and Pile-CC (Figure 3), ensuring its applicability to natural text.
> >
> > We will revise the manuscript to clarify how GINC captures meaningful properties of natural language while complementing our experiments on real-world data. Thank you for your constructive feedback.
> >
> > [2] An Explanation of In-context Learning as Implicit Bayesian Inference
> > Sang Michael Xie, Aditi Raghunathan, Percy Liang, Tengyu Ma

---

### Meta-Review · Area_Chair_Ra9F · 2024-12-21

**Metareview:**

This paper presents a metric to quantify the diversity of a text corpus with the aim to correlate it with quality of a language model trained on such a corpus. The relies on an embedding model called Task2Vec built on top of GPT2, and the diversity score is determined by calculating the expected cosine distance between Task2Vec representations of different batches of data.

Reviewers agree that this is an important problem to study as data diversity is an important factor for the performance of language models but the relationship between pre-training data diversity and downstream performance is currently poorly understood.

However, the empirical evidence for the correlation between diversity and performance is limited, particularly regarding the choice of datasets as they might not reflect realistic pretraining datasets, limited number of datasets, and potential confounders combining different datasets might have introduced. The reviewers also point out that the correlation between high diversity and the perplexity is inconsistent where in several cases, the language model performance declines with increased quality. Furthermore, the paper lacks a comparison with other diversity metrics or simpler embedding-based approaches. Based on the reviewers' criticism, I recommend reject at this time.

**Additional Comments On Reviewer Discussion:**

The authors acknowledge many of the limitations pointed out by the reviewers and promise to update them in the future revision of the paper. The authors have also provided justification for using limited number of datasets, lack of consistency in performance, and lack of comparison to other diversity metrics. The reviewers have also suggested comparing performance on downstream tasks which the authors argue is outside of the scope of this paper (and I agree with the authors on this front). Overall, the reviewers have chosen not to update their scores. I agree with some of the justifications provided by the authors but the limitations such as inconsistency of results, lack of more datasets, as they stand are not easily addressable by writing changes and require substantial changes.

---

### Decision · Program_Chairs · 2025-01-22

Reject